# DIFFERENTIABLE GRADIENT SAMPLING FOR LEARNING IMPLICIT 3D SCENE RECONSTRUCTIONS FROM A SINGLE IMAGE

**Shizhan Zhu, Sayna Ebrahimi, Angjoo Kanazawa, Trevor Darrell**
UC Berkeley

## ABSTRACT

Implicit neural shape functions, e.g. occupancy fields or signed distance functions, are promising 3D representations for modeling arbitrary 3D surfaces. However, existing approaches that use these representations for single-view 3D reconstruction require 3D supervision signals at *every* location in the scene, posing difficulties when extending to real-world scenes where ideal watertight geometry necessary to compute dense supervision is difficult to obtain. In such cases, constraints on the spatial gradient of the implicit field, rather than the value itself, can provide a training signal, but this has not been employed as a source of supervision for single-view reconstruction in part due to the difficulties of differentiably sampling a spatial gradient from a feature map. In this paper, we derive a novel closed-form Differentiable Gradient Sampling (DGS) solution that enables backpropagation of the loss on spatial gradients to the feature maps, thus allowing training on large-scale scenes without dense 3D supervision. As a result, we demonstrate single view implicit 3D surface reconstructions on real-world scenes via learning directly from a scanned dataset. Our model performs well when generalizing to unseen images from Pix3D or downloaded directly from the Internet (Fig. 1). Extensive quantitative analysis confirms that our proposed DGS module plays an essential role in our learning framework. Video and code are available at https://github.com/zhusz/ICLR22-DGS.

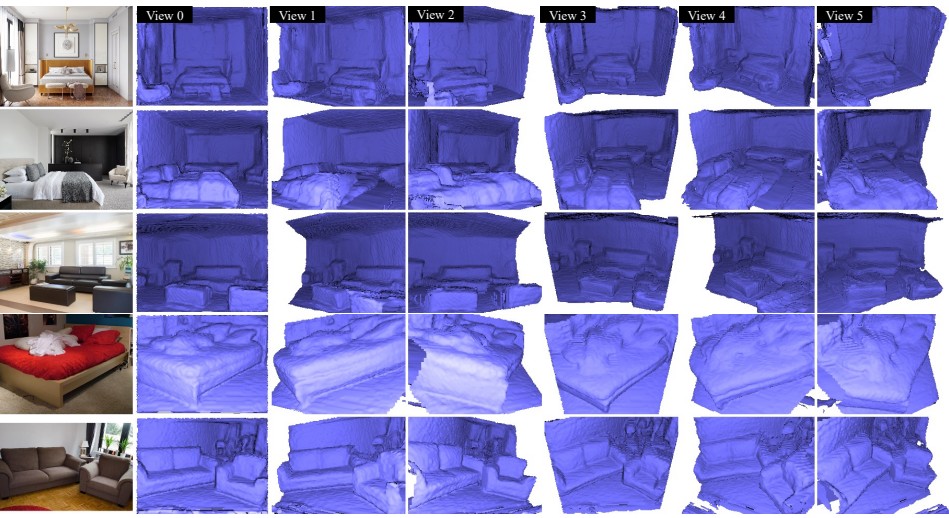

Figure 1: Given a single RGB image as the input (the first column), our model can predict its 3D implicit surface reconstruction (shown in six novel views in the last six columns). The test images for the first 3 rows are downloaded from the Internet and the last 2 rows are from the pix3d dataset.

# 1 INTRODUCTION

Recent studies have shown promising learning capabilities of implicit models for 3D geometry representations (Park et al., 2019; Mescheder et al., 2019; Mildenhall et al., 2020; Sitzmann et al., 2019; Gropp et al., 2020; Saito et al., 2019; Yu et al., 2020). Rather than explicitly representing 3D geometry with a textured mesh (Kato et al., 2018) or point cloud (Fan et al., 2017), implicit representations express a function of points in the space. Among many recent promising implicit models, the occupancy field (Mescheder et al., 2019; Popov et al., 2020; Chen & Zhang, 2019b) and the Signed Distance Field (SDF) (Park et al., 2019; Xu et al., 2019; Li & Zhang, 2021) are particularly suitable for high resolution topology free surface reconstruction. Given a query point $(x, y, z) \in \mathbb{R}^3$ in the space, the occupancy function returns the occupancy status of the point $o(x, y, z) \in \{+, -\}$, while the SDF value returns the closest distance from a point to the surface of the given 3D geometry, as well as its sign for its occupancy. The surface is defined by the classification boundary of the occupancy field or the zero level-set of the SDF (Lorensen & Cline, 1987).

Despite the strong representational power of these models and their success on predicting 3D objects from single image, learning to predict complex 3D geometry, such as scenes, from a real world image remains a challenge for this class of methods. Obtaining the occupancy label, or the sign, from noisy non-watertight meshes is non-trivial (Atzmon & Lipman, 2020) (Fig. 2). Furthermore, producing the label on the fly from a large number of triangle meshes, as encountered in typical real-world scenes with complex geometry, can be computationally prohibitive (Straub et al., 2019; Chang et al., 2017; Dai et al., 2017; Song et al., 2017). Storing dense query points sufficiently with pre-computed occupancies or SDF for complex geometries, or meta data of meshes like octrees or hashing, is a possibility, but incurs severe engineering and computational burdens on storage and runtime loading (Yu et al., 2021). The challenge is exacerbated for predicting scenes in particular, since unlike objects scenes in the view frustrum typically cannot be enclosed by a pre-defined range of space and do not have a categorical canonical coordinate system (Popov et al., 2020).

Recent studies on fitting a single scene geometry to partial 3D observations (Gropp et al., 2020; Sitzmann et al., 2020) have pointed out the value of the 3D spatial gradient of the implicit field. For instance, the gradients of the SDF on surface points are actually the surface normals and the 2-norm of the spatial gradients for any point in the space is 1, known as the Eikonal equation (Crandall & Lions, 1983). Similarly, in this work, we propose to regularize the gradient of the occupancy field to be zero away from the surface (Fig. 3(c)). Thus, one can train the full occupancy field or SDF with only the surface point clouds, thanks to these constraints on the gradients of the implicit shape models, even when the value of the occupancy or the SDF itself is not available everywhere. However, existing approaches (Gropp et al., 2020; Sitzmann et al., 2020) have only demonstrated this benefit in the cases of geometry inputs only, where a network is fit to a particular scene per model without any conditioning on input images. When locally sampled image features are used in the feed-forward prediction, it is necessary to derive a differentiable gradient sampling solution over the feature pixel map.

In this paper, we tackle this challenge by deriving a closed-form differentiable gradient sampling (DGS) solution for learning a single-view 3D implicit reconstruction model (Fig. 4). Our novel propagation of the loss gradient of the spatial gradient back to the feature maps (Eq. 6) serves in addition to the existing spatial gradient sampling (Eq. 5) and loss gradient back-propagation (Eq. 4) used in existing deep learning frameworks (Paszke et al., 2019; Abadi et al., 2016). The resulting end-to-end learning scheme with supervision over the spatial gradients opens the potential to train a model generalizable to unseen test cases of complex scenes with only surface point cloud supervision — a typical setting for real-world data — without requiring ground truth per-query-point labels.

Our contributions include a framework to propagate spatial gradients through a spatial feature sampling procedure, and a novel closed-form DGS solution. Experiments on real-scanned data (ScannetV2 (Dai et al., 2017)) shows that DGS enables training a single-image 3D implicit reconstruction model that can generalize to unseen scenes, with only imperfect surface annotations as the supervision (e.g. non-watertight meshes). Experiments on synthetic data (ShapeNet Chang et al. (2015)) indicate that our learning framework without using dense per-query-point training labels demonstrates competitive performance compared to the oracle scenario where dense occupancy labels are available. To the best of our knowledge, DGS-enabled shape inference provides the first single-view implicit shape reconstruction on real scene datasets which can generalize accurately to unseen scenes from different datasets or domains (see Figure 1).

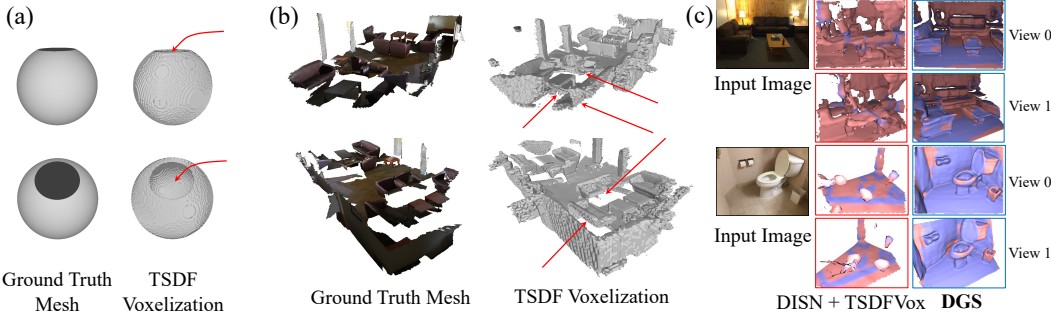

Figure 2: (a, b) Truncated SDF (TSDF) (Curless & Levoy, 1996) Voxelization results of the non-watertight ground truth meshes (each shown in two views). (a) is a simple sphere and (b) is a real scene from the ScannetV2 training data. After depth-fusion and internal-filling (Popov et al., 2020), the inside space of both geometries (a, b) remains empty (red arrows), causing severe noise for training the occupancy field or the SDF prediction model. This type of noise particularly affects the single-view prediction problem, as no additional predicted depth surface from other views are available. (c) As a result of learning the implicit prediction directly from the inaccurate and low-resolution TSDF voxels (due to engineering constraints on runtime loading and memory bottleneck for the sufficiently dense pre-computed query point occupancy labels), the prediction result (DISN + TSDFVox) is clearly inferior compared to our results (DGS). The surface color denotes the evaluation of the "precision", with the larger blue region, the higher "precision".

## 2 RELATED WORKS

**3D Implicit Representations** Among all the 3D representations, implicit models are advantageous for arbitrarily high resolution modeling (unlike voxels with fixed resolution and no detailed surface modeling) and easy learning (unlike meshes which assume a fixed topology). Implicit models typically learn to establish a mapping between a query point and the prediction of the point. Groueix et al. proposed to learn a mapping from the uv texture map to the 3D surface point. More recent works (Park et al., 2019; Mescheder et al., 2019; Xu et al., 2019; Popov et al., 2020; Chen & Zhang, 2019a) focus on mapping the query point coordinates to the signed distance field or the occupancy field. In addition to these pure geometry modeling, recent works like (Saito et al., 2019) also model the surface texture via learning a mapping from the surface point to the RGB value, or use the RGB loss as the supervision signal (Niemeyer et al., 2020; Yariv et al., 2020). In contrast to defining textures explicitly on

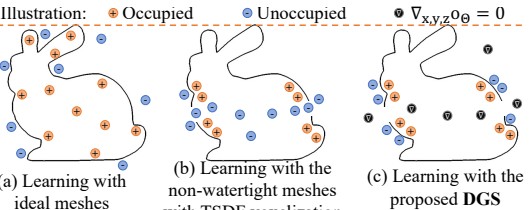

Figure 3: Illustration of the loss imposition for the occupancy prediction scenario. (a) When learning from the ideal mesh for ShapeNet objects, we can directly supervise the training with the accurate occupancy labels. (b) On scans of real scenes with imperfections (Fig. 2(b)), the TSDF voxelization produces severe noise for training. Specifically, a considerable fraction of the objects are "empty" inside. (c) Our learning scheme with DGS alleviates these issues via enabling imposition of losses on the gradients all the way back to image features.

the surface, Mildenhall et al. (2020) proposed the volumetric rendering representation which maps the point coordinate and the viewing angle to volumetric textures. Oechsle et al. (2021); Wang et al. (2021); Yariv et al. (2021) use volume renderings for fitting high quality geometry of single scenes, but these are not generalizable to unseen scenes. In multi-view stereo setting, Murez et al. (2020); Sun et al. (2021); Božič et al. (2021) proposed to accumulate the multi-view predictions in a TSDF voxel volume. We believe our proposed DGS module is necessary when extending most of the existing implicit surface representations into feed-forward models that are conditioned on input image(s), and training these models with a loss on the gradient of the predicted field. In this work, we focus on 3D geometry reconstruction reconstructions from single view, as an example of the application of the proposed DGS module.

**Differentiable Operations** End-to-end deep learning requires all modules in a computation path to be differentiable. Many differentiable modules are proposed to serve a particular functionality. The

spatial transformer layer (Jaderberg et al., 2015) is one of the early approaches for differentiable sampling of the feature map. Kato et al. (2018) proposed to render the mesh into image in a differentiable manner to optimize the mesh or the camera parameters to fit to a particular image. A similar idea has also been applied for rendering point clouds (Wiles et al., 2020; Yifan et al., 2019). Tulsiani et al. (2017) and Mildenhall et al. (2020) proposed to compute the ray rendering in an accumulating manner. One common feature of these differentiable operations is a strategy to soften the existing explicit modeling (Liu et al., 2019), and replace the gradient of the loss with approximate signals, making sure those signals provide guidance in the right direction. Our differentiable gradient sampler also seeks to soften the spatial gradient between the adjacent pixels.

## 3 LEARNING FRAMEWORK

**Problem definition and notations.** We aim to train a feed-forward deep model for predicting the 3D implicit surface reconstruction conditioned on a single image. We denote the input RGB image as $I \in \mathbb{R}^{m \times n \times 3}$. Like most feed-forward models, our model employs an image encoder. We denote the extracted 2D features as $\phi$. Our model $\hat{f}_\Theta(\cdot)$, parameterized by $\Theta$, takes in the image feature $\phi$ and predicts for each 3D location $(x, y, z)$ the implicit value $\hat{f}_\Theta(\phi; x, y, z)$. We denote the ground truth value as $f(\phi; x, y, z)$. For the occupancy field, the implicit field value $f$ represents the occupancy probability. We denote the predicted occupancy probability as $\hat{o}_\Theta(\phi; x, y, z) \in [0, 1]$ and the ground truth binary occupancy label as $o(\phi; x, y, z) \in \{ " - ", " + " \}$, where "$-$" represents unoccupied space and "$+$" vice versa. For SDF, the implicit field value $f$ represents the signed distance between the query point and its projection on the surface. We denote the predicted and the ground truth signed distance as $\hat{s}_\Theta(\phi; x, y, z) \in \mathbb{R}$ and $s(\phi; x, y, z) \in \mathbb{R}$ respectively.

**Background.** Typical fully supervised training (Xu et al., 2019; Popov et al., 2020) imposes the loss to each individual sampled query points by comparing the predicted field value $\hat{f}_\Theta(\phi; x, y, z)$ with the ground truth value $f(\phi; x, y, z)$. For occupancy predictions, we compute the loss as $\sum_{\{I, \mathcal{P}_I\} \in \mathcal{D}} \sum_{(x,y,z) \in \mathcal{P}_I} \text{BCE}(\hat{o}_\Theta(\phi; x, y, z), o(\phi; x, y, z))$ (Fig. 3(a)), where "BCE" represents the binary cross entropy loss, $\mathcal{D}$ represents the whole single-view image training set, and $\mathcal{P}_I$ represents the set of all the possible query point sampling locations $(x, y, z)$ within the view frustum. For SDF, we compute the loss as $\sum_{\{I, \mathcal{P}_I\} \in \mathcal{D}} \sum_{(x,y,z) \in \mathcal{P}_I} |\hat{s}_\Theta(\phi; x, y, z) - s(\phi; x, y, z)|$.

### 3.1 PROPOSED FRAMEWORK

As obtaining the full labels $f(\phi; x, y, z)$ for query points $(x, y, z)$ from every location in $\mathcal{P}_I$ is non-trivial for complex geometry from real-world (Fig. 2) and due to additional engineering constraints on runtime loading and memory bottleneck for the sufficiently dense pre-computed query point occupancy labels, we propose to incorporate the loss w.r.t. the spatial gradients. For occupancy of the implicit field,

$$
\begin{aligned}
\mathcal{L} = \sum_{\{I, \mathcal{P}_I\} \in \mathcal{D}} ( \sum_{(x,y,z) \in \mathcal{P}_I - \mathcal{P}_I^0} \lambda_{or} \| \nabla_{x,y,z} \hat{o}_\Theta(\phi; x, y, z) \| \\
+ \sum_{(x,y,z) \in \mathcal{P}_I^{0+}} \text{BCE}(\hat{o}_\Theta(\phi; x, y, z), " + ") + \sum_{(x,y,z) \in \mathcal{P}_I^{0-}} \text{BCE}(\hat{o}_\Theta(\phi; x, y, z), " - "))
\end{aligned}
\tag{1}
$$

where $\nabla_{x,y,z} \hat{o}_\Theta(\phi; x, y, z) = [\nabla_x \hat{o}_\Theta(\phi; x, y, z), \nabla_y \hat{o}_\Theta(\phi; x, y, z), \nabla_z \hat{o}_\Theta(\phi; x, y, z)]^\top$ denotes the spatial gradient of the occupancy prediction, $\mathcal{P}_I^{0+}$ and $\mathcal{P}_I^{0-}$ denote inward and outward near-surface query points, and $\lambda_{or}$ represents the loss weight of the occupancy geometric regularization. The facing direction of the mesh surface (determining inward or outward) can be determined by normals (if available), or via rendering the surface in all views as in the RGBD captures (Dai et al., 2017) (with the surface facing toward camera as the "outward" side). The three terms in Eq. 1 correspond to the three types of losses in Fig. 3(c) ("$\nabla$", "$+$", "$-$") respectively. For SDF, we set the loss similar to Gropp et al. (2020):

$$
\begin{aligned}
\mathcal{L} = \sum_{\{I, \mathcal{P}_I\} \in \mathcal{D}} ( \sum_{(x,y,z) \in \mathcal{P}_I} \lambda_{sr} |\| \nabla_{x,y,z} \hat{s}_\Theta(\phi; x, y, z) \|_2 - 1| + \sum_{(x,y,z) \in \mathcal{P}_I^0} |\hat{s}_\Theta(\phi; x, y, z)| \\
+ \sum_{(x,y,z) \in \mathcal{P}_I^0} \lambda_{sn} \| \nabla_{x,y,z} \hat{s}_\Theta(\phi; x, y, z) - \nabla_{x,y,z} s(\phi; x, y, z) \|)
\end{aligned}
\tag{2}
$$

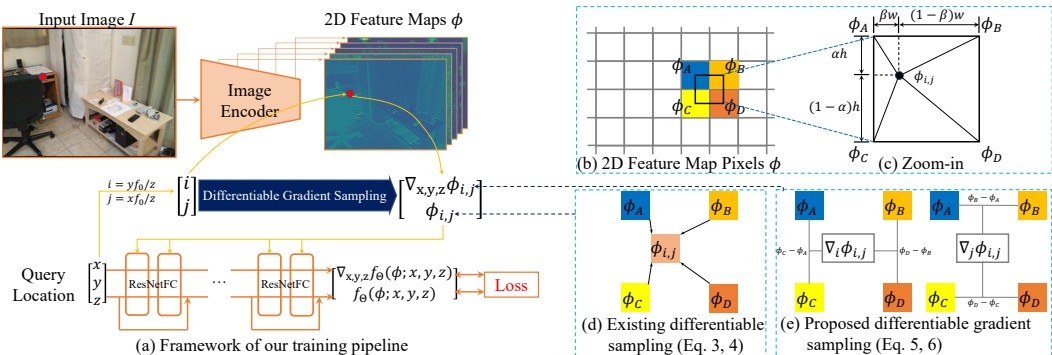

Figure 4: Overview of our learning framework (a) and differentiable gradient sampling (b, c, d, e).

where $\mathcal{P}_I^0$ denotes the query points on the ground truth surface only, and $\lambda_{sr}$ and $\lambda_{sn}$ represents the loss weight for the signed distance geometric regularization term and normal term respectively. The three terms are the Eikonal regularization (Crandall & Lions, 1983), surface zero-SDF loss, and the surface normal loss respectively, where the last term is optional (Gropp et al., 2020). Note now the loss functions in Eq. 1, 2 no longer require the implicit ground truth label for query points far away from the ground truth surface, enabling training with the real-world imperfect scanned data. In both cases, the availability of surface normals is optional.

Practically, we found only the proposed loss for the occupancy model (Fig. 3(c), Eq. 1) to be effective in the real-world single-view feed-forward scenario, and use Eq. 1 rather than Eq. 2 in all of our experiments. There are a few reasons. First, learning with the Eikonal term (Eq. 2) suffers severely from its sensitivity of model initialization. We found either the SDF or the Truncated SDF (TSDF) representation model cannot easily converge during training. Specifically, we found it poses numerical difficulties when the model is learned to predict a large distance value or a constant truncation value for the majority of the query points in the air that is far from any surfaces. This becomes a major problem when we depart from the single objects scenario (Mescheder et al., 2019; Popov et al., 2020) or single scene fitting (Gropp et al., 2020) to large schale scenes, which is our focus. Second, our loss function (Eq. 1) can significantly save memory footprint during training and enable large batch size training, which is crucial in our learning framework. Unlike in Eq. 2 where all the query points require spatial gradient computation, which leads to $3\times$ higher of memory footprint, in our loss function (Eq. 1), only the non-surface query points do. We set the query point batch size of the non-surface points (512 in practice) to be much smaller than the critical near-surface points (4096 in practice), enabling learning with the image batch size of 32 in our real-scene training.

## 3.2 SAMPLING WITH DIFFERENTIABLE GRADIENTS

A distinct difference compared to existing works is that our spatial gradient $\nabla_{x,y,z}\hat{f}_\Theta(\phi; x, y, z)$ is also conditioned on the pixels in $\phi$. Gropp et al. (2020) devised the model $f_\Theta$ to fit to a single scene, and the spatial gradient $\nabla_{x,y,z}\hat{f}_\Theta(x, y, z)$ can be conveniently computed because it does not involve the image sampling procedure. Jiang et al. (2020); Lin et al. (2020) used a non-spatial global feature for inference and hence bypassed 2D sampling. In our learning framework, the spatial gradient computation must undergo the sampling procedure.

We name our gradient computation involving the sampling operation as the *Pixel Conditioned Gradients* and derive a closed-form solution, *Differentiable Gradient Sampling* (DGS), for handling forward and backward propagation. Figure 4(a) provides an illustration of our training pipeline. Each layer in our network tracks both the response of the layer as well as its spatial gradient w.r.t. $(x, y, z)$. While it is well established to track the layer-wise spatial gradient for fully-connected (FC) layers or convolutions in existing works (Gropp et al., 2020), tracking the spatial gradients and back-propagating the loss to the feature map pixels $\phi$ through the sampling module has not been studied. To this end, we derive the closed-form sampling scheme for tracking and propagating the spatial gradients $\nabla_{x,y,z}\phi = [\nabla_x\phi, \nabla_y\phi, \nabla_z\phi]^\top$ through the sampling layer.

**Background - 2D Differentiable Sampling.** Differentiably sampling pixel values from a grid of 2D feature map with the given pixel locations $(i, j)$ is a common operation. Throughout our paper, we define the pixel coordinates in the normalized coordinate system that ranges from -1 to 1. As

illustrated in Fig. 4(d), given the feature map $\phi$ and the sampling locations $(i, j)$, the resulting sampled value $\phi(i, j)$ is

$$\phi(i, j) = (1 - \alpha)((1 - \beta)\phi_A + \beta\phi_B) + \alpha((1 - \beta)\phi_C + \beta\phi_D) \tag{3}$$

Please refer to Fig. 4(c) for the definitions of $\alpha, \beta$ and $\phi_A, \phi_B, \phi_C, \phi_D$. Without loss of generality we use bilinear interpolation. During training, the gradient from the loss can be back-propagated via

$$\frac{\partial L}{\partial \phi_A} = (1 - \alpha)(1 - \beta)\frac{\partial L}{\partial \phi(i, j)}. \tag{4}$$

Equation 4 is for Pixel A and similarly to the other 3 pixels (please refer Eq. 9).

**2D Differentiable Gradient Sampling.** Our learning framework (Sec. 3.1, Fig. 4(a)) requires the extension of the sampling capability from just the feature value response $\phi_{i,j}$ to its spatial gradient $\nabla_{i,j}\phi(i, j) = [\nabla_i\phi(i, j), \nabla_j\phi(i, j)]^\top$. During the forward and the backward propagation, both the sampled feature response $\phi_{i,j}$ and its spatial gradient $\nabla_{i,j}\phi(i, j)$ are recorded for further propagation (Fig. 4(e)):

$$\nabla_i\phi(i, j) = \frac{(1 - \beta)(\phi_C - \phi_A) + \beta(\phi_D - \phi_B)}{h}. \tag{5}$$

Please refer to Eq. 10 for $\nabla_j\phi(i, j)$. Hence, during the forward pass, we compute the spatial gradient via Eq. 5 in addition to the existing value sampling (Eq. 3). During the backward pass, we compute the loss gradient over the spatial gradient via

$$
\begin{aligned}
\frac{\partial L}{\partial \phi_A} &= \frac{\partial L}{\partial \phi(i, j)}\frac{\partial \phi(i, j)}{\partial \phi_A} + \frac{\partial L}{\partial \nabla_i\phi(i, j)}\frac{\partial \nabla_i\phi(i, j)}{\partial \phi_A} + \frac{\partial L}{\partial \nabla_j\phi(i, j)}\frac{\partial \nabla_j\phi(i, j)}{\partial \phi_A} \\
&= (1 - \alpha)(1 - \beta)\frac{\partial L}{\partial \phi(i, j)} + (-\frac{1 - \beta}{h})\frac{\partial L}{\partial \nabla_i\phi(i, j)} + (-\frac{1 - \alpha}{w})\frac{\partial L}{\partial \nabla_j\phi(i, j)},
\end{aligned}
\tag{6}
$$

where $w$ and $h$ are the width and height of a pixel in normalized coordinate system, s.t. $w = 2/W$ and $h = 2/H$ for the feature map with the size $W \times H$. Please refer to Eq. 11 for $\frac{\partial L}{\partial \phi_B}, \frac{\partial L}{\partial \phi_C}$ and $\frac{\partial L}{\partial \phi_D}$.

**3D Differentiable Gradient Sampling.** We now extend the sampling to 3D. We model the camera as the pin-hole camera. For any point $(x, y, z)$ in the camera space, we seek for its projected 2D locations $(i, j)$ based on the focal length $f_0$ via $i = \frac{yf_0}{z}, j = \frac{xf_0}{z}$. The feed forward pass is

$$
\begin{aligned}
\nabla_x\phi(x, y, z) &= \nabla_i\phi(i, j) \cdot \frac{\partial i}{\partial x} + \nabla_j\phi(i, j) \cdot \frac{\partial j}{\partial x} = \nabla_j\phi(i, j) \cdot \frac{f_0}{z}, \\
\nabla_y\phi(x, y, z) &= \nabla_i\phi(i, j) \cdot \frac{\partial i}{\partial y} + \nabla_j\phi(i, j) \cdot \frac{\partial j}{\partial y} = \nabla_i\phi(i, j) \cdot \frac{f_0}{z}, \\
\nabla_z\phi(x, y, z) &= \nabla_i\phi(i, j) \cdot \frac{\partial i}{\partial z} + \nabla_j\phi(i, j) \cdot \frac{\partial j}{\partial z} = \nabla_i\phi(i, j) \cdot (-\frac{yf_0}{z^2}) + \nabla_j\phi(i, j) \cdot (-\frac{xf_0}{z^2}).
\end{aligned}
\tag{7}
$$

Note that here, our notation of the sampled 2D image feature $\phi(x, y, z)$ refers to the same as $\phi(i, j)$, as $(i, j)$ is the projected pixel coordinates of $(x, y, z)$ that represent the exact projected 2D locations when extracting the 2D features.

During the backward propagation procedure, the DGS accumulates the gradient via

$$
\begin{aligned}
\frac{\partial L}{\partial \phi_A} &= \frac{\partial L}{\partial \phi(x, y, z)}\frac{\partial \phi(x, y, z)}{\partial \phi_A} + \frac{\partial L}{\partial \nabla_x\phi(x, y, z)}\frac{\partial \nabla_x\phi(x, y, z)}{\partial \phi_A} \\
&\quad + \frac{\partial L}{\partial \nabla_y\phi(x, y, z)}\frac{\partial \nabla_y\phi(x, y, z)}{\partial \phi_A} + \frac{\partial L}{\partial \nabla_z\phi(x, y, z)}\frac{\partial \nabla_z\phi(x, y, z)}{\partial \phi_A} \\
&= \frac{\partial L}{\partial \phi(x, y, z)} \cdot (1 - \alpha)(1 - \beta) + \frac{\partial L}{\partial \nabla_x\phi(x, y, z)} \cdot (-\frac{1 - \alpha}{w}) \cdot \frac{f_0}{z} \\
&\quad + \frac{\partial L}{\partial \nabla_y\phi(x, y, z)} \cdot (-\frac{1 - \beta}{h}) \cdot \frac{f_0}{z} + \frac{\partial L}{\partial \nabla_z\phi(x, y, z)} \cdot (\frac{1 - \beta}{h} \cdot \frac{yf_0}{z^2} + \frac{1 - \alpha}{w} \cdot \frac{xf_0}{z^2}).
\end{aligned}
\tag{8}
$$

Please refer to Eq. 12 for $\frac{\partial L}{\partial \phi_B}, \frac{\partial L}{\partial \phi_C}$, and $\frac{\partial L}{\partial \phi_D}$.

Table 1: Intersection over Union % (IoU ↑) benchmarking result on the high-realism ShapeNet. Our approach (the last 4 rows) demonstrates competitive performance compared to state-of-the-art approaches (the top 4 rows) even trained without the dense occupancy labels as used in the oracle settings of these existing works. Our comparisons with the state-of-the-art approaches are direct ablations as we maintain exactly the same experimental setups except for the loss function. In addition, our approach also comfortably outperforms the ablation baselines (the middle 5 approaches).

| Category | Craft | Rifle | Disp. | Lamp | Spk. | Box | Chair | Bench | Car | Plane | Sofa | Table | Phone | Mean |
|---|---|---|---|---|---|---|---|---|---|---|---|---|---|---|
| OccNet | 49.6 | 39.7 | 49.7 | 33.3 | 49.3 | 42.1 | 42.8 | 30.9 | 57.2 | 41.7 | 60.7 | 42.4 | 64.8 | 46.5 |
| DISN | 54.5 | 52.5 | 50.2 | 39.2 | 53.3 | 46.0 | 50.6 | 37.1 | 58.6 | 48.5 | 64.9 | 48.4 | 67.6 | 51.6 |
| CoReNet | 60.5 | 67.5 | 61.0 | 46.9 | 56.8 | 51.3 | 59.7 | 47.1 | 61.1 | 58.4 | 68.7 | 56.9 | 77.3 | 59.5 |
| DISN(Res50)+DVR | 61.1 | 64.5 | 61.9 | 46.9 | 58.2 | 54.4 | 59.6 | 48.0 | 59.4 | 58.4 | 69.5 | 57.2 | 78.7 | 59.8 |
| Abl.-NoGrad | 11.9 | 4.9 | 23.6 | 15.4 | 31.4 | 30.6 | 25.1 | 10.1 | 20.8 | 7.7 | 28.3 | 20.1 | 21.2 | 19.3 |
| Abl.-NoGrad (10 %) | 23.7 | 19.9 | 33.1 | 26.5 | 40.0 | 35.2 | 35.6 | 18.6 | 33.0 | 15.7 | 38.8 | 30.7 | 41.0 | 30.1 |
| Abl.-NoGrad (30 %) | 26.4 | 15.5 | 36.8 | 23.6 | 40.6 | 36.0 | 36.8 | 19.5 | 38.5 | 19.1 | 47.8 | 30.9 | 39.4 | 31.6 |
| Abl.-NoGrad (50 %) | 39.7 | 30.0 | 39.7 | 28.4 | 45.0 | 37.5 | 39.0 | 23.6 | 50.6 | 29.0 | 53.5 | 36.2 | 49.7 | 38.6 |
| Abl.-FixedE | 49.6 | 52.5 | 44.4 | 33.0 | 46.3 | 39.9 | 43.4 | 27.5 | 57.5 | 46.9 | 58.8 | 39.1 | 68.0 | 46.7 |
| OccNet w/ Eq. 1 | 50.7 | 42.6 | 50.4 | 33.0 | 50.1 | 43.7 | 44.4 | 32.9 | 56.8 | 41.6 | 60.9 | 44.0 | 68.4 | 47.7 |
| DISN w/ Eq. 1 | 53.3 | 51.2 | 51.9 | 38.7 | 52.1 | 43.8 | 50.8 | 36.2 | 58.8 | 47.4 | 64.6 | 47.4 | 66.6 | 51.0 |
| CoReNet w/ Eq. 1 (DGS) | 61.1 | 67.5 | 62.7 | 44.2 | 54.8 | 49.6 | 59.5 | 45.4 | 59.4 | 59.9 | 69.8 | 55.1 | 78.0 | 59.0 |
| DISN(Res50)+DVR w/ Eq. 1 (DGS Best) | 60.8 | 62.6 | 62.3 | 47.1 | 57.7 | 53.5 | 59.8 | 47.4 | 59.2 | 58.6 | 70.7 | 57.4 | 77.0 | 59.6 |

## 4 EXPERIMENTS

### 4.1 LEARNING FROM SYNTHETIC DATA (SHAPENET)

**Dataset.** Following Popov et al. (2020), we conduct experiments on ShapeNet with the low-realism and the high-realsim renderings. For low-realism, we use the renderings of various models from Choy et al. (2016). Following Mescheder et al. (2019), we split the dataset into the training set of 30661 models, the validation set of 4371 models, and the test set of 8751 models. During testing, we follow Mescheder et al. (2019) to only test the first rendering of each model. Similar to all the prior works, we use the same 13 classes to report any performance via grouping all the test cases into the same class. For high-realism, we follow the split and evaluation protocol of the single-object scenes Popov et al. (2020) - we only use the 13 categories used by Choy et al. (2016), filter out repeated samples, and finally construct the data with 666,565 models from the training set, 96,084 from the validation set, and 189,748 from the test set. Following Popov et al. (2020), we only evaluate the first 1% test cases of the test set (1898 samples).

**Metrics.** We use the Intersection-of-Union (IoU) for evaluating the model performances. For low-realism renderings, we follow the protocol of IoU benchmarking from Mescheder et al. (2019) that evaluating occupancy of the specified 100K query points in the space. For high-realism renderings, we follow the protocol from Popov et al. (2020) to evaluate occupancy prediction of the $128 \times 128 \times 128$ grid. Following Popov et al. (2020), the grid cube is a unit cube that spans from $-0.5$ to $0.5$ for $x$ and $y$, and from $f/2$ to $1 + f/2$ for $z$ in the camera coordinate system.

**Comparison with state-of-the-art approaches.** On high-realism ShapeNet, we build our model on top of the representative state-of-the-art models - OccNet (Mescheder et al., 2019), DISN (Xu et al., 2019), CoReNet (Popov et al., 2020) for their 3D volumetric implicit modeling capability. We also incorporate a strong baseline named "DISN (ResNet50) + DVR (Niemeyer et al., 2020)", where we replace the originally used VGG (Simonyan & Zisserman, 2014) with ResNet-50 as the image encoder as used in CoReNet (Popov et al., 2020), and devised the 5-layer fully connected ResNet as the decoder as used in DVR (Niemeyer et al., 2020). In each DGS experiment, we maintain exactly the same encoder and decoder architecture, optimization parameters (e.g. the learning rate and the epsilon of Adam Kingma & Ba (2014)) and the prediction format (per-query occupancy probability). We set $\lambda_{or}$ to be 0.01 in our loss function (Eq. 1). Note that the DGS experiments can only access nearsurface training signals, in contrast to the oracle learning setting as in the state-of-the-art approaches where models were trained with dense occupancy labels. We name the DGS version of the CoReNet model as the **DGS** model, and the DISN+DVR counterpart as the **DGS-Best** model. Quantitative results are reported in Tab. 4 for the low-realism evaluation setting, and in Tab. 1 for the high-realism setting[1]. Our competitive experimental results indicate that our learning

---

[1]We noticed that in our original submission version of the paper, our experiments were conducted with evaluating the $64 \times 64 \times 64$ volume grids due to an error. We thus revised our results in Tab. 1 with the evaluation results returned from the $128 \times 128 \times 128$ volume grid, and attach our previous results in Tab. 3. Please refer to Sec. A for detailed explanation.

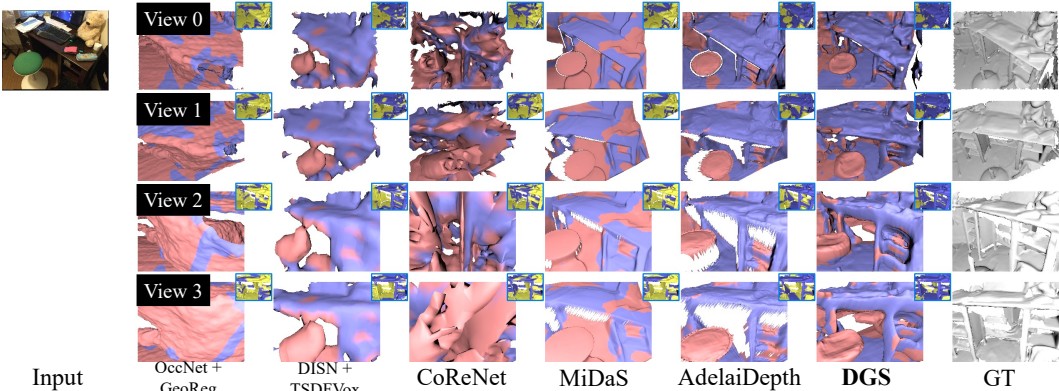

Figure 5: Qualitative comparisons on one challenging test case on ScannetV2. For each predicted surface with red and sky-blue colors, sky-blue indicates "positive precision" for that surface region, while red indicates "negative precision". The ground truth surface is shown on the top-right corner of each prediction with gold and navy-blue colors, navy-blue indicates "positive recall", while gold indicates "negative recall". The larger the blue region is, the higher the F1 score would be.

Table 2: Benchmarking results of single view 3D surface reconstruction on ScannetV2 test set.

|  | Acc (↓) | Compl (↓) | Chamfer (↓) | Prec (↑) | Recall (↑) | F1-score (↑) |
|---|---|---|---|---|---|---|
| OccNet + GeoReg | 18.3 | 18.5 | 18.4 | 30.4 | 28.5 | 26.8 |
| DISN + TSDFVox | 13.0 | 53.3 | 33.1 | 25.3 | 10.2 | 12.4 |
| CoReNet | 19.5 | 15.6 | 17.6 | 30.9 | 36.3 | 29.9 |
| MiDaS | 13.9 | 18.5 | 16.2 | 41.9 | 30.1 | 34.4 |
| AdelaiDepth | **9.7** | 18.8 | 14.2 | **46.4** | 33.2 | 37.9 |
| Ablation-NoGrad | 17.0 | 11.7 | 14.3 | 34.1 | 47.0 | 36.5 |
| Ablation-FixedE | 15.0 | 20.3 | 17.7 | 31.6 | 29.8 | 27.3 |
| **DGS** | 14.3 | **11.5** | **12.9** | 39.7 | **49.4** | **41.6** |

framework along with the differentiable gradient sampling layer implementation plays the critical role when learning with only the near-surface training labels, and can achieve similar performance without the dense occupancy training labels.

Please refer to Sec. A for ablation baselines details and low-realism ShapeNet experimental settings.

## 4.2 LEARNING FROM REAL SCANNED DATASETS (SCANNETV2)

**Dataset.** We use ScannetV2 (Dai et al., 2017) for training and evaluating the performance of the models on the real images. We follow the standard training / testing split as used in Sun et al. (2021) and Murez et al. (2020), where 1513 scenes are for trainval (with 1201 for training and 312 for validation), and 100 for testing. Each scene is provided with multiple image capture as well as the associated camera pose. We train the models with all the views given in the training / validation set (2423872 frames in total, after filtering out frames with invalid extrinsic poses), while for testing, we select 10 frames with different extrinsic poses for each test scene. Practically, since all the frames of the scenes are in the form of video clips, with adjacent frames associated with similar extrinsic cameras poses, we select the 10 frames for each frame via extracting every 100 frames from each scene video (e.g. Frame 1, 101, 201, ..., 901), resulting in 1000 frames in total in our test set (10 frames per scene, with 100 test scenes).

**Metrics.** We use the same evaluation metrics following Sun et al. (2021) and Murez et al. (2020). Since we are the first, to our knowledge, to evaluate single view 3D implicit reconstruction on the ScannetV2 benchmark, and here we only evaluate the geometries within the camera view frustum rather than the whole scene geometries as in Sun et al. (2021); Murez et al. (2020). In addition, we also only evaluate the geometries in front of the *amodal* depth for each pixel ray, where only the space in front of the wall, bounded within the ceiling and the floor, is evaluated. We define the amodal depth for a pixel ray to be delineated by the minimum between the closest structure-category surface (e.g. walls and doors, etc) and the farthest surface. In practice, in order to accommodate the evaluation of the surfaces right on the amodal depth, we slack the evaluation scope with a factor $\lambda$ (1.05 in our case) multiplied with the amodal depth. This evaluation protocol would be equivalent to the "single-layered" protocol used in Sun et al. (2021), within our single-view scenario.

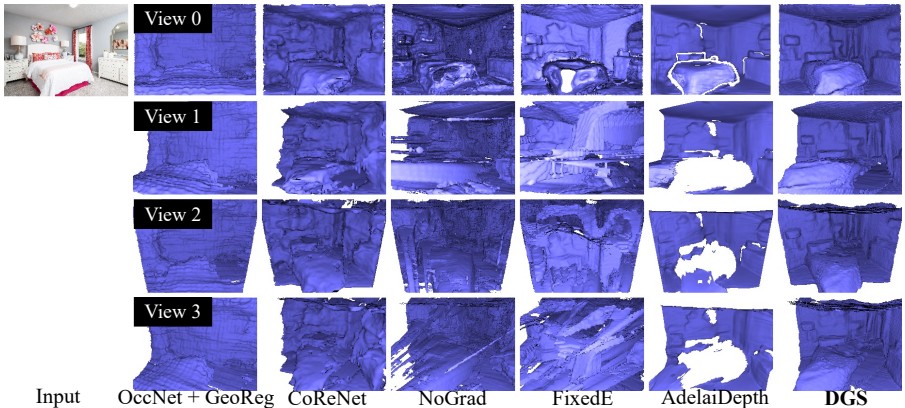

Figure 6: Qualitative comparisons on an unseen test image downloaded from the internet.

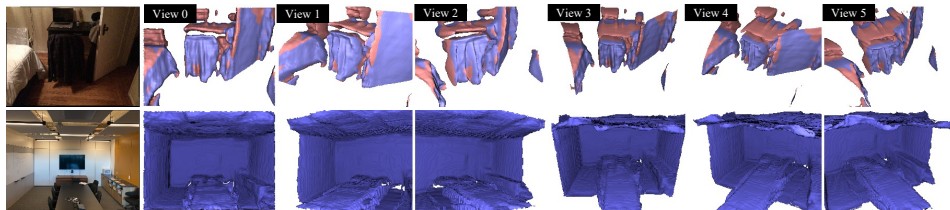

Figure 7: Two representative failure cases of our approach.

Due to the inherent ambiguity of the scaling and shifting of the predicted 3D single view geometry (Ranftl et al., 2019; Yin et al., 2021), we follow them by computing the best scale and shift comparing the predicted depth with the ground truth depth. For approaches that predict 3D surfaces, the rendered depth map from the predicted mesh would be used for calculating the scale and shift.

Please refer to Sec. B for details on baselines.

**Results.** We provide quantitative comparison in Tab. 2 and qualitative comparison in Fig. 5 respectively. Our model outperforms all state-of-the-art approaches as well as the ablation models. Compared to the synthetic data scenario in Tab. 1 and 4, our model demonstrates large advantages when compared to existing state-of-the-art approaches, as our motivation stems from addressing learning from imperfect 3D labels directly from the real scan data. Compared to single-view depth prediction approaches trained on massive data (Lasinger et al., 2019; Yin et al., 2021), our approach does not prevail on "Acc" and "Prec". This is due to the fact that these two metrics only project the predicted surface to the ground truth surface, giving advantages to approaches that only predict the visible surface. Our approach still prevails for other metrics (Chamfer Distance and F1), which are considered as the most important metrics (Božič et al., 2021; Sun et al., 2021; Murez et al., 2020).

**Generalization to Unseen Scenes (Pix3d and Open-Domain Images).** We further test our model without further finetuning directly on unseen scenes for evaluating the generalizability of the learned model. We run our model on Pix3d Sun et al. (2018) as well as test images downloaded directly from the internet. We provide a detailed qualitative comparison in Fig. 6 and more results in Fig. 1. The results further indicate our learning framework exhibits promise for unseen scene generalization.

**Failure Cases.** We provide two representative failure cases of our approach in Fig. 7. The first case (the first row) demonstrates difficulties in predicting the floor occupancy as a result of the noisy and non-watertight mesh during training. The second case (the second row) shows that our model cannot identify small objects (e.g. chairs) and not predict the invisible partition of these objects.

## 5 CONCLUSIONS

We have presented our learning framework for real-world 3D implicit surface reconstruction from a single view image. Owing to our unique learning framework that directly trains the model from the raw scan data and our novel occupancy loss function over the gradients, we are able to go beyond the existing works for single-objects reconstruction or single view fitting. Thanks to our differentiable gradient sampling module we enable efficient and memory-efficient end-to-end training from images and demonstrated single view 3D surface reconstruction results on scenes for the first time, to our knowledge.

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

# APPENDIX

In this document, we provide additional experimental details and results on ShapeNet in Sec. A, additional qualitative results on ScannetV2 in Sec. B, analysis of numerical gradient approximation in Sec. C, additional results generalizing to external unseen scenes in Sec. D and E, as well as the details of the formulation and derivations of DGS in Sec. F.

## A  ADDITIONAL DETAILS AND RESULTS ON SHAPENET

**Additional Quantitative Results.** We provide our original results on high-realism ShapeNet in Tab. 3. These results are evaluated with the $64 \times 64 \times 64$ volume grids due to an error in our prior code base. Based on the results in both Tab. 1 and Tab. 3, we found our proposed learning framework performs competitively with the oracle training setting, even though our approach does not utilize dense occupancy labels as used in the oracle state-of-the-art approaches (OccNet, DISN, CoReNet and DISN (ResNet50) + DVR). On the other hand, we observe that our model demonstrates slight improvements when evaluated with the resolution of $64 \times 64 \times 64$, while evaluating the same models with the resolution of $128 \times 128 \times 128$ would reverse the course. In particular, we found categories like "car" exhibited major performance decrease on all the approaches with the higher resolution evaluating setting. This might be due to the fact that these categories demonstrate empty ground truth inside the objects, and and at a higher resolution, the metrics are biased to penalize models that predict "occupie" inside the object. However, the competitive performance in both resolution settings indicates that our learning with only the surface labels does not deteriorate the performance, and can be faithfully applied toward the real-scene settings where the dense occupancy labels are truly unavailable.

Table 3: Intersection over Union % (IoU ↑) benchmarking result on the high-realism ShapeNet with the resolution of $64 \times 64 \times 64$. Please refer to Tab. 1 (the resolution of $128 \times 128 \times 128$) for details.

| Category | Craft | Rifle | Disp. | Lamp | Spk. | Box | Chair | Bench | Car | Plane | Sofa | Table | Phone | Mean |
|---|---|---|---|---|---|---|---|---|---|---|---|---|---|---|
| OccNet | 54.7 | 48.7 | 55.5 | 38.6 | 57.1 | 52.0 | 48.9 | 39.7 | 70.6 | 48.9 | 63.6 | 49.2 | 71.7 | 53.8 |
| DISN | 60.9 | 63.5 | 56.9 | 46.4 | 61.5 | 55.2 | 57.4 | 46.7 | 72.5 | 57.2 | 68.2 | 55.7 | 74.5 | 59.7 |
| CoReNet | 62.5 | 65.5 | 63.2 | 48.2 | 63.0 | 56.7 | 60.6 | 48.3 | 73.7 | 58.1 | 69.8 | 55.0 | 75.1 | 61.5 |
| DISN(Res50)+DVR | 66.9 | 74.4 | 67.3 | 54.5 | 66.1 | 63.2 | 66.2 | 58.6 | 71.2 | 67.2 | 72.5 | 63.1 | 82.6 | 67.2 |
| Abl.-NoGrad | 14.1 | 6.1 | 27.3 | 18.1 | 37.3 | 40.4 | 28.7 | 13.0 | 26.2 | 9.6 | 30.5 | 23.4 | 25.1 | 23.1 |
| Abl.-NoGrad (10 %) | 28.2 | 25.8 | 39.3 | 32.4 | 48.2 | 46.5 | 42.0 | 24.6 | 42.4 | 20.0 | 42.7 | 37.1 | 49.4 | 36.8 |
| Abl.-NoGrad (30 %) | 34.1 | 25.1 | 46.6 | 30.8 | 49.8 | 47.9 | 45.1 | 28.1 | 51.4 | 26.5 | 54.9 | 38.0 | 52.7 | 40.8 |
| Abl.-NoGrad (50 %) | 47.8 | 41.5 | 47.6 | 35.1 | 54.1 | 49.2 | 46.8 | 32.1 | 66.0 | 37.9 | 59.6 | 44.4 | 60.2 | 47.9 |
| Abl.-FixedE | 52.6 | 57.0 | 47.8 | 37.4 | 52.6 | 49.7 | 47.4 | 33.0 | 69.2 | 51.0 | 60.2 | 42.9 | 69.2 | 51.5 |
| OccNet w/ Eq. 1 | 56.8 | 52.6 | 56.8 | 39.3 | 58.1 | 53.6 | 51.1 | 42.4 | 71.7 | 49.7 | 65.5 | 50.9 | 76.0 | 55.7 |
| DISN w/ Eq. 1 | 60.4 | 63.9 | 59.6 | 46.2 | 61.1 | 55.8 | 58.3 | 46.6 | 73.2 | 57.3 | 68.7 | 55.5 | 76.7 | 60.2 |
| CoReNet w/ Eq. 1 (DGS) | 63.3 | 70.4 | 65.7 | 49.0 | 61.2 | 57.0 | 62.1 | 50.9 | 70.1 | 62.8 | 70.3 | 56.9 | 78.3 | 62.9 |
| DISN(Res50)+DVR w/ Eq. 1 (DGS Best) | 66.8 | 75.1 | 68.7 | 55.1 | 65.7 | 63.7 | 66.5 | 58.2 | 71.5 | 67.4 | 72.9 | 63.5 | 84.2 | 67.6 |

**Ablation Study Details.** We conduct ablation studies to further validate the importance of derived DGS module, via attempting work-around training methods without DGS.

*i) NoGrad* - To test the performance of the baseline when only the surface data are provided, we train this ablation model in exactly the same way compared to its original model, with the only exceptions that only the near-surface points are equipped with training labels, and we do not use any training labels from the non-surface query points (where it is not necessary to backpropagate the loss gradient of the spatial gradient using DGS). To further evaluate how the rate of known voxels affects the learning performance, we enlarge the near-surface region and evaluate when the rate is 10%, 30% and 50%. An increase in the performances among these baselines would indicate the importance of knowing more voxel labels if our proposed gradient loss (Eq. 1) is not imposed.

*ii) FixedE* - We train with both the near-surface as well as the spatial gradient supervision, without DGS - meaning the loss gradient of the spatial gradient would not back-propagate to any module before the sampling module - in our case, the feature encoder network. Note all the other losses without gradient sampling can still back-propagated to the feature maps.

We report the ablation results in Tab. 1. Both experiments are conducted with the high-realism ShapeNet data. We comfortably outperform all the in-house ablation baselines, validating the essential roles of DGS in our learning framework.

**Experiments on low-realism ShapeNet.** For our low-realism evaluation setting, all the baseline approaches reported their results in the papers. For OccNet and DISN, the reported results are based on knowing the category canonical view prior, which demonstrates considerable privilege with respect to accuracy Popov et al. (2020). Hence, we mark the results from the literature as OccNet-Privilege and DISN-Privilege (OccNet-Priv. and DISN-Priv. for short). Compared to their privileged setting, our results demonstrate superior results even without the category canonical view prior privilege. To further provide the the baseline results where these two approaches are without the category canonical view prior, we retrain their models with the released codes, and report the results in the "OccNet" and "DISN" row respectively. The results further provide evidence that the category canonical view prior demonstrates privilege on accuracy, as observed in Popov et al. (2020).

**Qualitative Results.** We provide qualitative results in Fig. 9-13 as the additional illustration of the performance of all the approaches on ShapeNet.

Table 4: Intersection over Union % (IoU ↑) benchmarking result on the low-realism ShapeNet. Our proposed DGS learning advances state-of-the-art approaches (OccNet, DISN, CoReNet and D$^2$IM-Net) compared to the reported performance from the literatures.

| Category | Craft | Rifle | Disp. | Lamp | Spk. | Box | Chair | Bench | Car | Plane | Sofa | Table | Phone | Mean |
|---|---|---|---|---|---|---|---|---|---|---|---|---|---|---|
| OccNet-Priv. | 53.0 | 47.4 | 47.1 | 37.1 | 64.7 | **73.3** | 50.1 | 48.5 | 73.7 | 57.1 | 68.0 | 50.6 | 72.0 | 57.1 |
| DISN-Priv. | 60.2 | 68.0 | 57.7 | 39.7 | 55.9 | 53.1 | 54.9 | 54.2 | 77.0 | 61.7 | 67.1 | 48.9 | 73.6 | 59.4 |
| OccNet | 49.9 | 48.0 | 55.4 | 39.5 | 57.0 | 43.8 | 58.5 | 45.1 | 54.0 | 45.8 | 68.0 | 50.7 | 68.3 | 52.6 |
| DISN | 49.0 | 44.4 | 55.5 | 39.0 | **67.3** | 71.7 | 49.0 | 41.2 | 64.7 | 45.0 | 66.4 | 50.7 | 70.5 | 55.0 |
| CoReNet | 54.0 | 64.6 | 57.2 | 42.1 | 60.8 | 50.9 | **63.0** | 50.8 | 57.3 | 53.0 | **70.6** | 55.5 | 73.1 | 57.9 |
| D$^2$IM-Net | **63.4** | **68.1** | 52.7 | 42.1 | 51.8 | 48.6 | 56.1 | **55.0** | **79.8** | 55.8 | 65.4 | 53.7 | 76.2 | 59.1 |
| **DGS** | 57.0 | 65.7 | **58.6** | **45.5** | 58.6 | 55.2 | 59.8 | 48.2 | 71.6 | **56.8** | 68.1 | **55.6** | **78.4** | **60.0** |

# B  ADDITIONAL RESULTS ON SCANNETV2

**Baselines.** Since we demonstrate the first attempt to predict 3D implicit surface of scenes from a single image, very few exiting works provide a direct baseline performance to our task. For Occ-Net (Mescheder et al., 2019), since its image feature extraction is not local, and its gradient propagation does not require sampling, we use its direct application with Gropp et al. (2020) "OccNet + GeoReg" as one baseline. We train DISN (Xu et al., 2019) with the TSDF voxelization labels (Murez et al., 2020; Sun et al., 2021; Božič et al., 2021). For CoReNet (Popov et al., 2020), we stick to its own voxelization and internal filling toolbox for obtaining the training labels. Since CoReNet can only predict geometries within the fixed range of space, we tried our best to pick the best cube location based on the dataset statistics. We also incorporate depth approaches (Ranftl et al., 2019; Yin et al., 2021) for comparison by finetuning their weights on ScannetV2. Lastly, we compare with the two ablation models *NoGrad* and *FixedE* as introduced in Sec. 4.1.

**Implementation details for DGS.** Since the DISN variant architectures (Xu et al., 2019; Niemeyer et al., 2020; Yu et al., 2020) demonstrate higher degress of flexibility of representing any point in the space, in contrast to 3D convolutions that are fixed for a particular range of space (Popov et al., 2020), we build our model based on the former architectures. Similar to Niemeyer et al. (2020); Yu et al. (2020), we use 5 residual blocks in the fully connected part with the first 3 blocks receiving the 2D features and its spatial gradients. Our image encoder uses the same RexNext101 architecture (Xie et al., 2017) as in Yin et al. (2021) and starts the training with the pre-trained weights. We use a batch size of 32 images with 2048 near-surface points and 512 non-surface points per image. Similar to the ShapeNet experiments, we set $\lambda_{or}$ to be 0.01. We found in the real-world setting, this parameter setting is crucial for the convergence of the learning procedure.

**Quantitative Results for 2.5D evaluation metrics.** We provide the quantitative 2.5D evaluation results in Tab. 5. Our approach constantly outperforms all other implicit based baselines. While our approach falls short slightly when compared to the existing depth-based approaches, we claim that our approach is not directly trained with the massive depth training data. Our results indicate that our depth performance is still on par with the state-of-the-art depth prediciton approaches.

**Additional qualitative results.** We provide additional qualitative results on ScannetV2 Fig. 14-18.

Table 5: Benchmarking results of the depth metrics on ScannetV2.

|  | AbsRel ($\downarrow$) | AbsDiff ($\downarrow$) | SqRel ($\downarrow$) | RMSE ($\downarrow$) | LogRMSE ($\downarrow$) | $\delta_1$ ($\uparrow$) | $\delta_2$ ($\uparrow$) | $\delta_3$ ($\uparrow$) |
|---|---|---|---|---|---|---|---|---|
| OccNet + GeoReg | 13.5 | 23.7 | 7.2 | 33.2 | 17.3 | 82.2 | 95.5 | 99.0 |
| DISN + TSDFVox | 15.8 | 26.9 | 9.2 | 34.9 | 18.8 | 77.3 | 94.1 | 98.5 |
| CoReNet | 12.6 | 21.4 | 6.6 | 30.4 | 16.3 | 84.5 | 96.0 | 98.9 |
| MiDaS | 9.3 | 15.8 | 3.1 | 21.2 | 11.7 | 91.4 | 98.7 | 99.8 |
| AdelaiDepth | 6.1 | 10.5 | 1.9 | 15.7 | 8.6 | 95.3 | 99.1 | 99.8 |
| Ablation-Fair | 10.5 | 17.6 | 5.3 | 26.6 | 14.9 | 88.2 | 96.4 | 98.9 |
| Ablation-FixedE | 14.4 | 24.5 | 7.6 | 32.9 | 17.9 | 81.4 | 95.4 | 98.9 |
| **DGS** | 8.9 | 14.7 | 4.1 | 22.7 | 12.7 | 91.1 | 97.5 | 99.3 |

## C  ANALYSIS WITH NUMERICAL GRADIENT APPROXIMATION

As part of our proposed learning framework with the loss imposed on the spatial gradient, the differentiable gradient sampling module (Sec. 3.2) has a numerical alternative where we can perturb the query points to get its simulated numerical gradients. We further compare the closed-form performance with the numerical counterpart in Tab. 6. Please find qualitative comparison in Fig. 9-13.

We provide a closer look at our comparison to the simulated gradient baseline. We found in our experiments that the simulated gradient baseline demonstrates relatively severe convergence difficulties. As shown in Fig. 8, the simulated gradient model (red) does not observe loss drop in the first 10k training iterations. Despite its subsequent loss drop, which indicates the simulated gradient model can still learn to predict the geometry in moderate accuracy, its converged training loss is still higher than our DGS (blue). This aligns with our benchmarking evaluations in Tab. 6 that the closed-form variant (DGS) achieve better performance.

Table 6: Intersection over Union % (IoU $\uparrow$) benchmarking comparison between the closed-form solution and the numerical gradients on the high-realism ShapeNet. We report the performance comparison with both the resolution settings of $64 \times 64 \times 64$ and $128 \times 128 \times 128$ (Please refer to Sec. A for details).

| Category | Craft | Rifle | Disp. | Lamp | Spk. | Box | Chair | Bench | Car | Plane | Sofa | Table | Phone | Mean |
|---|---|---|---|---|---|---|---|---|---|---|---|---|---|---|
| Numerical (64) | 62.4 | 68.3 | 63.7 | 46.2 | 60.6 | 54.9 | 61.0 | 49.0 | **70.7** | 61.1 | 69.2 | 55.6 | 77.2 | 61.5 |
| **Closed-form (64)** | **63.3** | **70.4** | **65.7** | **49.0** | **61.2** | **57.0** | **62.1** | **50.9** | 70.1 | **62.8** | **70.3** | **56.9** | **78.3** | **62.9** |
| Numerical (128) | 59.7 | 66.9 | 61.3 | 41.8 | 54.1 | 48.5 | 58.2 | 43.8 | **59.8** | 59.0 | 68.5 | 54.3 | 77.0 | 57.9 |
| **Closed-form (128)** | **61.1** | **67.5** | **62.7** | **44.2** | **54.8** | **49.6** | **59.5** | **45.4** | 59.4 | **59.9** | **69.8** | **55.1** | **78.0** | **59.0** |

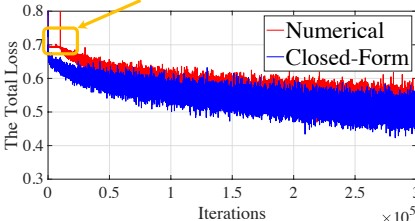

Figure 8: Convergence Analysis for the comparison between the closed-form (blue) and the numerical counterpart (red). Notably, the numerical counterpart does not observe loss drop in the first 10k iterations.

## D  ADDITIONAL GENERALIZABILITY QUALITATIVE RESULTS

We provide more qualitative results on our model generalizing to unseen images downloaded from the Internet in Fig. 19. Each result is visualized in 6 views. Once again, these additional visual results further indicate that our model demonstrates good generalizability capability for handling unseen indoor scene images.

Table 7: Benchmarking results of single view 3D surface reconstruction on Matterport3D test set (trained by the ScannetV2 dataset).

| | Acc (↓) | Compl (↓) | Chamfer (↓) | Prec (↑) | Recall (↑) | F1-score (↑) |
|---|---|---|---|---|---|---|
| OccNet + GeoReg | 24.8 | 33.3 | 29.0 | 21.9 | 27.7 | 23.4 |
| DISN + TSDFVox | 32.7 | 25.6 | 29.1 | 19.1 | 33.3 | 22.9 |
| CoReNet | 35.2 | 23.3 | 29.3 | 20.8 | 35.5 | 24.9 |
| AdelaiDepth | **14.1** | 32.9 | 23.5 | **33.7** | 28.7 | 29.9 |
| Ablation-NoGrad | 39.5 | **20.7** | 30.1 | 22.9 | **43.6** | 28.7 |
| Ablation-FixedE | 22.7 | 34.7 | 28.7 | 19.5 | 24.0 | 20.2 |
| **DGS** | 24.4 | 22.1 | **23.2** | 25.7 | 41.8 | **30.5** |

## E  EVALUATION OF THE GENERALIZABILITY TO MATTERPORT3D DATASET

To quantatively evaluate our generaizability to new datasets, we provide our results on the Matterport3D dataset (Chang et al., 2017) using our model trained from ScannetV2 as used in Sec. 4.2.

The Matterport3d dataset (Chang et al., 2017) collects indoor scenes of 90 full houses. Textured meshes are provided along with the scanned real images. Since the Matterport3D dataset does not provide an official train/test split, and our model is not trained on Matterport3d, we evaluate our model over all the 90 scenes in the dataset. In particular, we use the 129600 images (without the elevated pitch views where the camera is looking up at the ceiling) from the dataset, use the 10 first images from each scene (resulting in 900 test images in total), and test their reconstruction performance in the same way as we tested on ScannetV2. We provide the quantative evaluation result in Tab. 7. We can see from the table that the results align with our evaluation in Tab. 2 that our model demonstrates clear advantages. We also noticed that the AdelaiDepth Yin et al. (2021) baseline (finetuned on ScannetV2) performance is closer to our results (F1) score compared to our ScannetV2 evaluation. This is probably due to the fact that AdelaiDepth was directly pre-trained from a massive number of training images and potentially gives it some privilege when generalize to a new dataset.

## F  DETAILS OF THE FORMULATION AND DERIVATION OF DGS

In this section, we provide the full formulation as well as the derivation procedure of our DGS in this section as the supplementary to Sec. 3.2.

In Eq. 4, we provided the backward gradient for differentiable sampling for Pixel A. The backward gradient for Pixel B, C and D are

$$
\begin{aligned}
\frac{\partial L}{\partial \phi_B} &= (1 - \alpha)\beta \cdot \frac{\partial L}{\partial \phi(i, j)} \\
\frac{\partial L}{\partial \phi_C} &= \alpha(1 - \beta) \cdot \frac{\partial L}{\partial \phi(i, j)} \\
\frac{\partial L}{\partial \phi_D} &= \alpha\beta \cdot \frac{\partial L}{\partial \phi(i, j)}
\end{aligned}
\tag{9}
$$

In Eq. 5, we provided the forward gradient in the vertical pixel direction. The forward gradient of the horizontal pixel direction is

$$
\frac{\partial \phi(i, j)}{\partial j} = \frac{(1 - \alpha)(\phi_B - \phi_A) + \alpha(\phi_D - \phi_C)}{w}
\tag{10}
$$

In Eq. 6, we provided the backward gradient for only Pixel A. The backward gradient for Pixel B, C and D are

$$\frac{\partial L}{\partial \phi_B} = (1-\alpha)\beta \cdot \frac{\partial L}{\partial \phi(i,j)} + (-\frac{\beta}{h}) \cdot \frac{\partial L}{\partial \nabla_i \phi(i,j)} + \frac{1-\alpha}{w} \cdot \frac{\partial L}{\partial \nabla_j \phi(i,j)}$$

$$\frac{\partial L}{\partial \phi_C} = \alpha(1-\beta) \cdot \frac{\partial L}{\partial \phi(i,j)} + \frac{1-\beta}{h} \cdot \frac{\partial L}{\partial \nabla_i \phi(i,j)} + (-\frac{\alpha}{w}) \cdot \frac{\partial L}{\partial \nabla_j \phi(i,j)} \qquad (11)$$

$$\frac{\partial L}{\partial \phi_D} = \alpha\beta \cdot \frac{\partial L}{\partial \phi(i,j)} + \frac{\beta}{h} \cdot \frac{\partial L}{\partial \nabla_i \phi(i,j)} + \frac{\alpha}{w} \cdot \frac{\partial L}{\partial \nabla_j \phi(i,j)}$$

In Eq. 8, we provided the backward gradient for only Pixel A in the 3D setting. The backward gradient for Pixel B, C and D are

$$\frac{\partial L}{\partial \phi_B} = \frac{\partial L}{\partial \phi(x,y,z)} \cdot (1-\alpha)\beta + \frac{\partial L}{\partial \nabla_x \phi(x,y,z)} \cdot \frac{1-\alpha}{w} \cdot \frac{f_0}{z} + \frac{\partial L}{\partial \nabla_y \phi(x,y,z)} \cdot (-\frac{\beta}{h}) \cdot \frac{f_0}{z}$$

$$+ \frac{\partial L}{\partial \nabla_z \phi(x,y,z)} \cdot (\frac{\beta}{h} \cdot \frac{y f_0}{z^2} - \frac{1-\alpha}{w} \cdot \frac{x f_0}{z^2})$$

$$\frac{\partial L}{\partial \phi_C} = \frac{\partial L}{\partial \phi(x,y,z)} \cdot \alpha(1-\beta) + \frac{\partial L}{\partial \nabla_x \phi(x,y,z)} \cdot (-\frac{\alpha}{w}) \cdot \frac{f_0}{z} + \frac{\partial L}{\partial \nabla_y \phi(x,y,z)} \cdot \frac{1-\beta}{h} \cdot \frac{f_0}{z}$$

$$+ \frac{\partial L}{\partial \nabla_z \phi(x,y,z)} \cdot (-\frac{1-\beta}{h} \cdot \frac{y f_0}{z^2} + \frac{\alpha}{w} \cdot \frac{x f_0}{z^2})$$

$$\frac{\partial L}{\partial \phi_D} = \frac{\partial L}{\partial \phi(x,y,z)} \cdot \alpha\beta + \frac{\partial L}{\partial \nabla_x \phi(x,y,z)} \cdot \frac{\alpha}{w} \cdot \frac{f_0}{z} + \frac{\partial L}{\partial \nabla_y \phi(x,y,z)} \cdot \frac{\beta}{h} \cdot \frac{f_0}{z}$$

$$+ \frac{\partial L}{\partial \nabla_z \phi(x,y,z)} \cdot (-\frac{\beta}{h} \cdot \frac{y f_0}{z^2} - \frac{\alpha}{w} \cdot \frac{x f_0}{z^2})$$

$$(12)$$

As an illustration of the derivation for Eq. 8, we can obtain the backward gradient via

$$\frac{\partial L}{\partial \phi_A} = \frac{\partial L}{\partial \phi} \cdot \frac{\partial \phi}{\partial \phi_A} + \frac{\partial L}{\partial \nabla_x \phi} \cdot \frac{\partial \nabla_x \phi}{\partial \phi_A} + \frac{\partial L}{\partial \nabla_y \phi} \cdot \frac{\partial \nabla_y \phi}{\partial \phi_A} + \frac{\partial L}{\partial \nabla_z \phi} \cdot \frac{\partial \nabla_z \phi}{\partial \phi_A} \qquad (13)$$

The four partial derivatives with respect to $\phi_A$ in Eq. 13 can be determined based on the sampling rules. For example, based on Eq. 3, we can quickly obtain the first derivative with respect to $\phi_A$ via

$$\frac{\partial \phi}{\partial \phi_A} = (1-\alpha)(1-\beta) \qquad (14)$$

For the other three derivatives with respect to $\phi_A$ in Eq. 13, we can substitute Eq. 5, 10 into Eq. 7, and obtain them via

$$\frac{\partial \nabla_x \phi}{\partial \phi_A} = -\frac{f_0}{z} \cdot \frac{1-\alpha}{w}$$

$$\frac{\partial \nabla_y \phi}{\partial \phi_A} = -\frac{f_0}{z} \cdot \frac{1-\beta}{h} \qquad (15)$$

$$\frac{\partial \nabla_z \phi}{\partial \phi_A} = \frac{y f_0}{z^2} \cdot \frac{1-\beta}{h} + \frac{x f_0}{z^2} \cdot \frac{1-\alpha}{w}$$

We can then obtain Eq. 8 via substituting Eq. 14, 15 into Eq. 13.

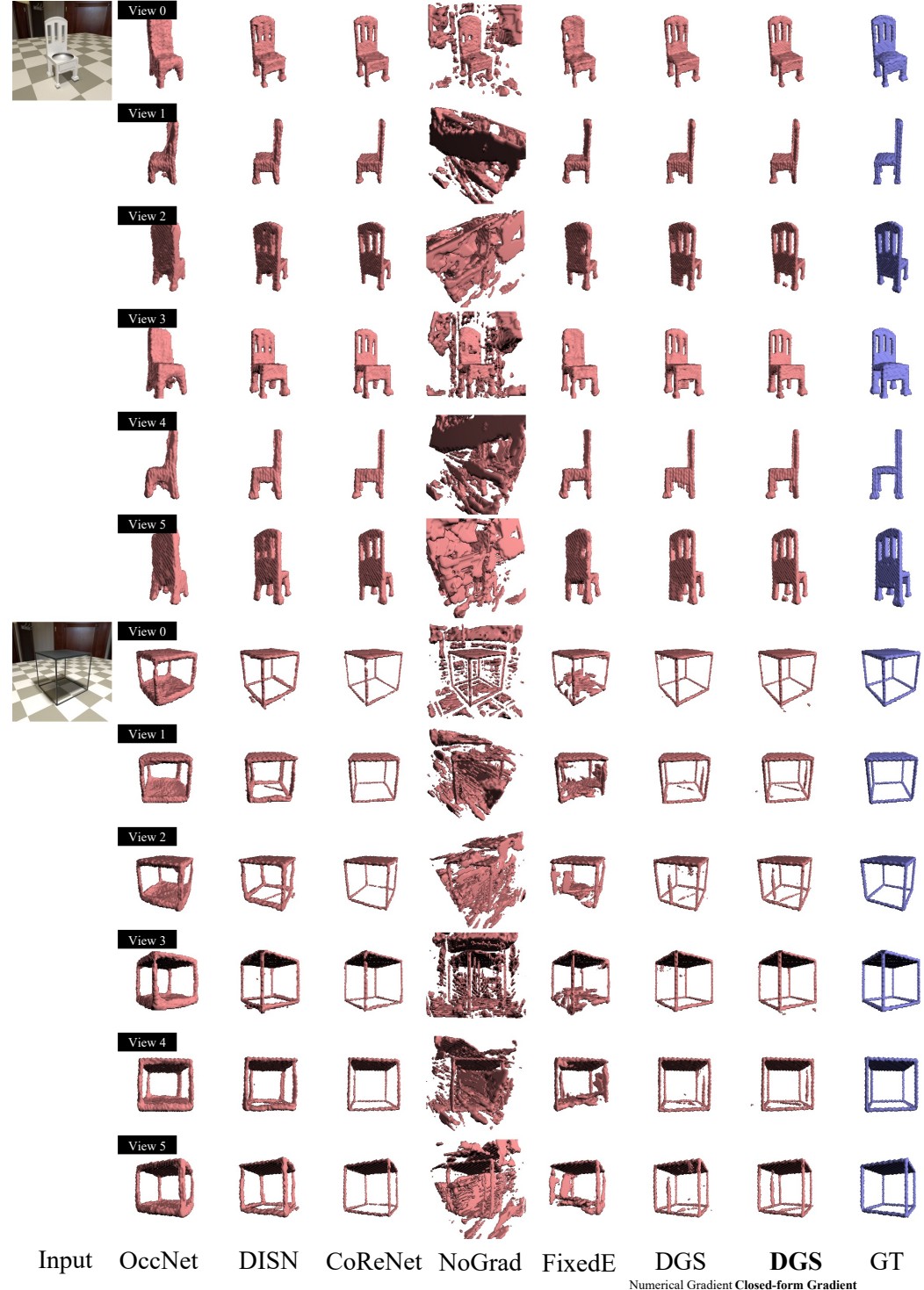

Figure 9: Quantitative comparison on the high-realism ShapeNet (without handpick: test case number 0 and 100). The reconstruction result of each approach is visualized in six different views, with the first view the same as the camera view, the first three views the same elevation as the camera view, and the last three view horizontal view.

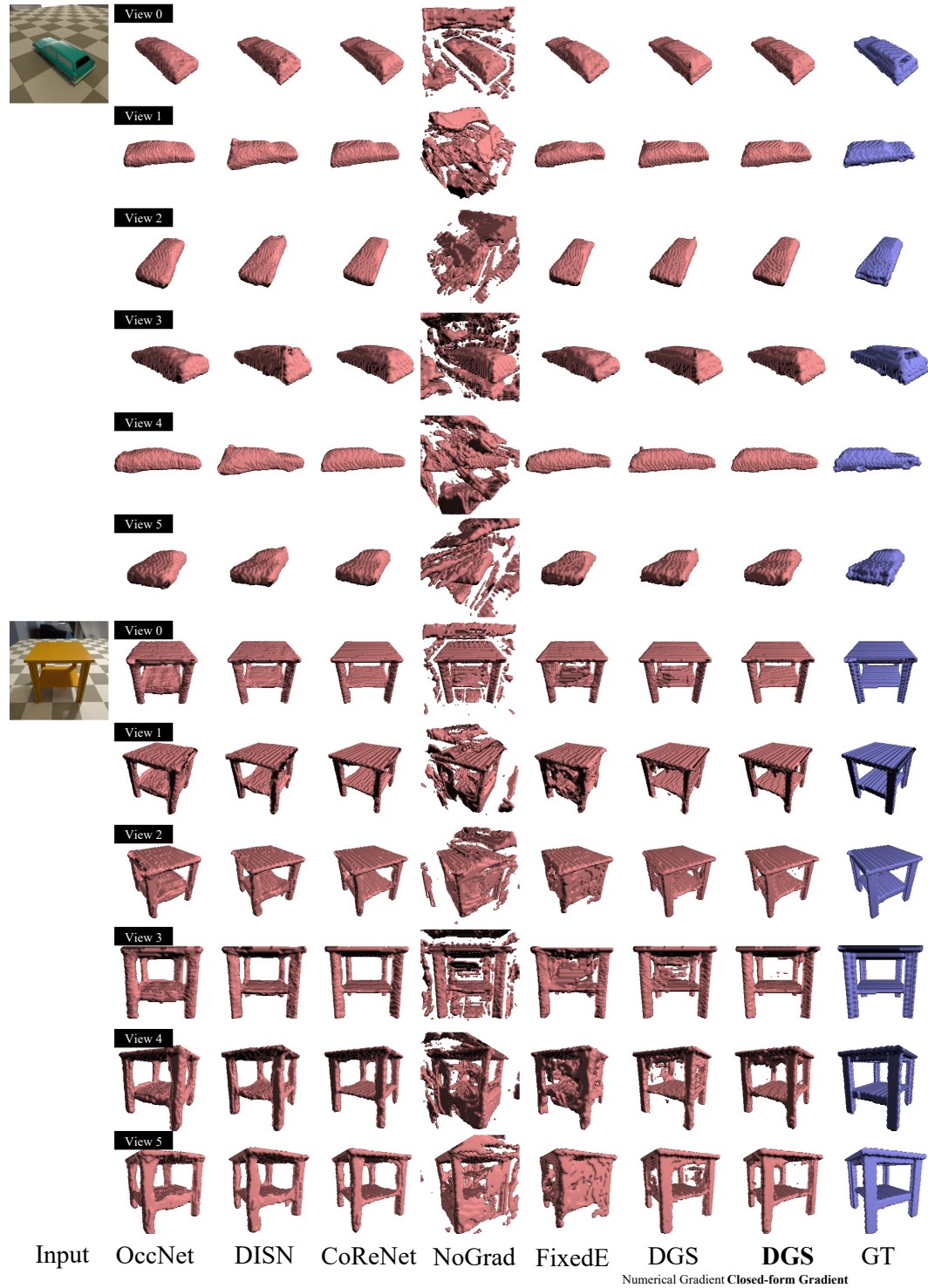

Input  OccNet  DISN  CoReNet  NoGrad  FixedE  DGS  **DGS**  GT

Numerical Gradient **Closed-form Gradient**

Figure 10: Quantitative comparison on the high-realism ShapeNet (without handpick: test case number 200 and 300). The reconstruction result of each approach is visualized in six different views, with the first view the same as the camera view, the first three views the same elevation as the camera view, and the last three view horizontal view.

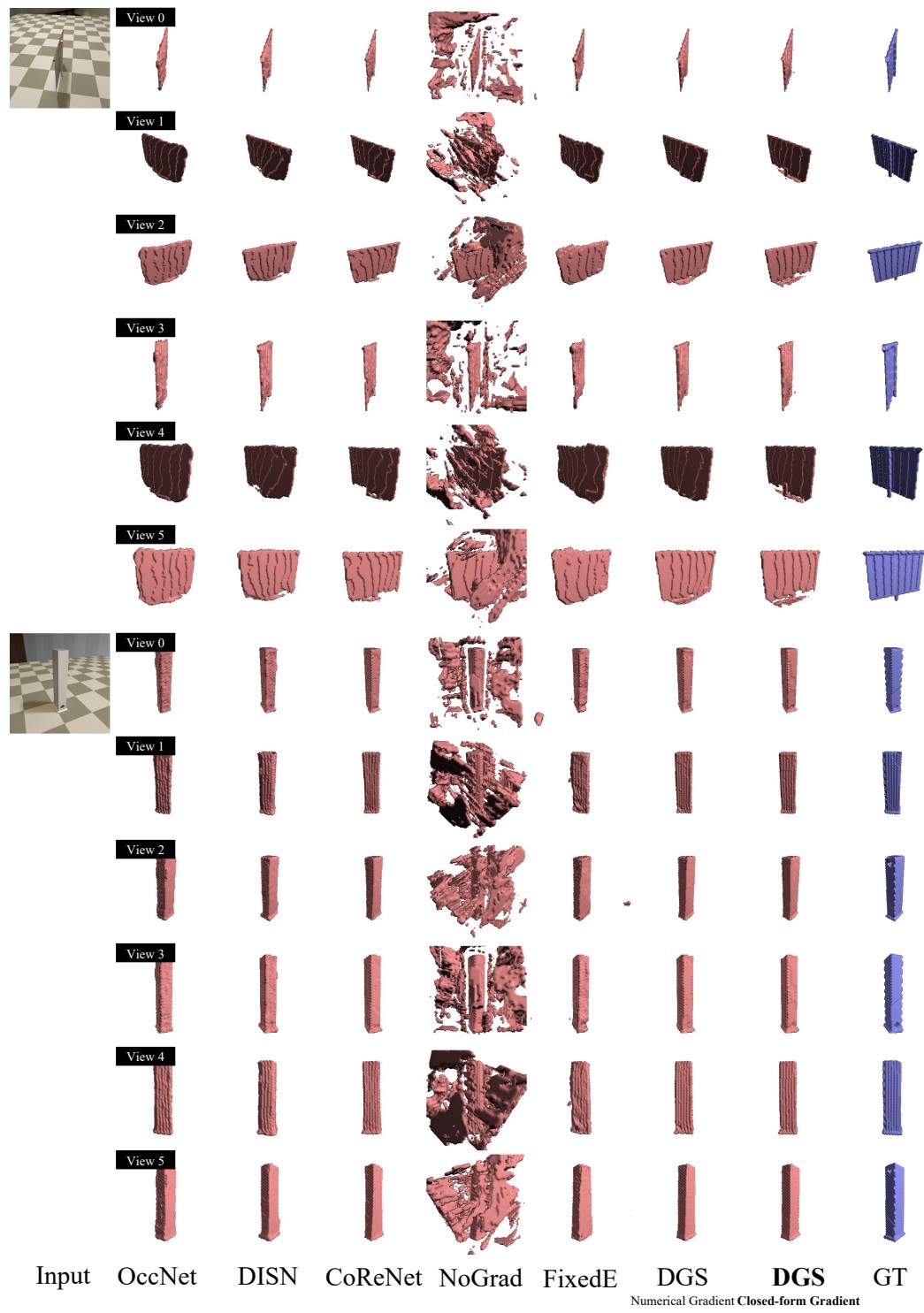

Figure 11: Quantitative comparison on the high-realism ShapeNet (without handpick: test case number 400 and 500). The reconstruction result of each approach is visualized in six different views, with the first view the same as the camera view, the first three views the same elevation as the camera view, and the last three view horizontal view.

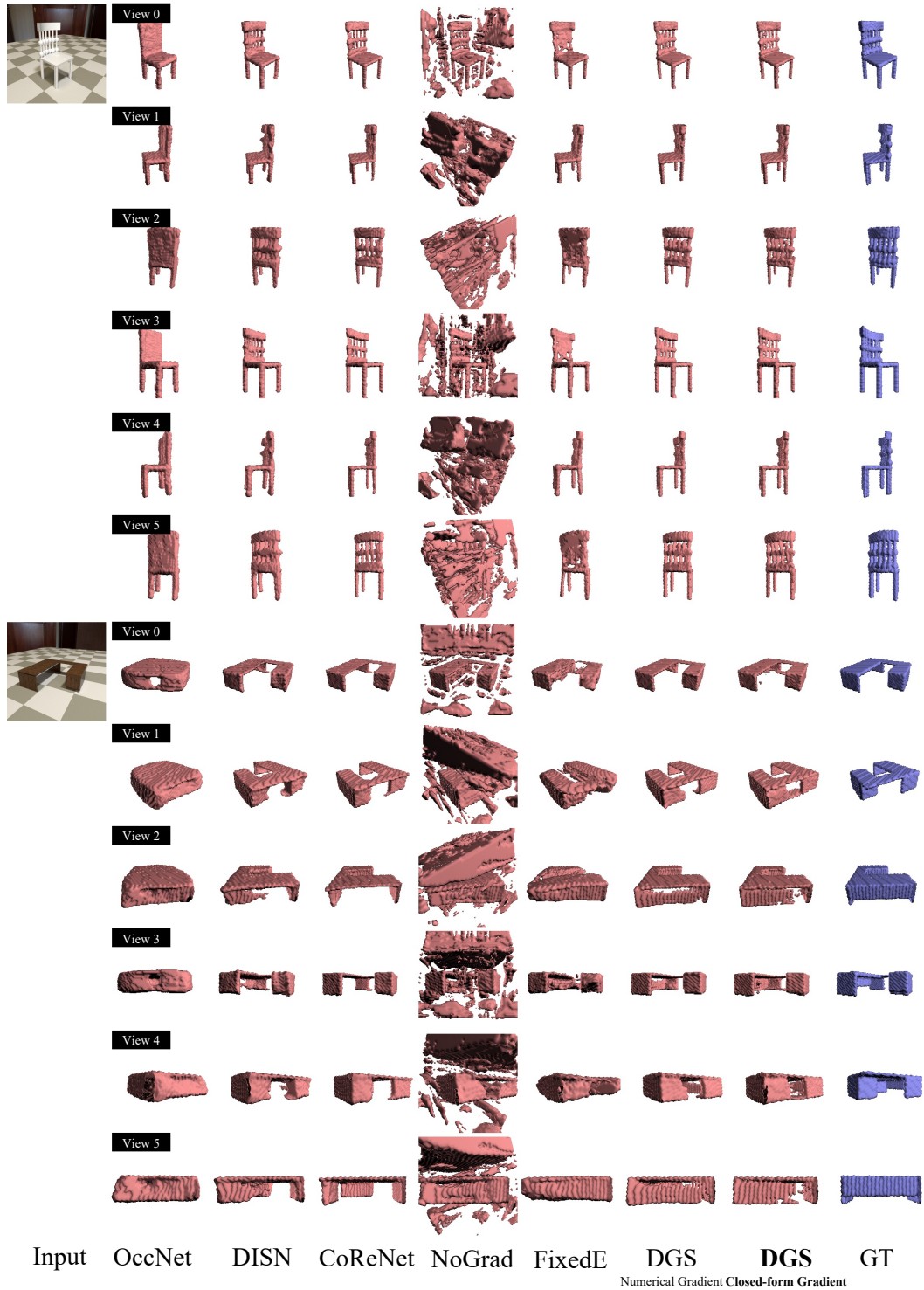

Input  OccNet  DISN  CoReNet  NoGrad  FixedE  DGS  **DGS**  GT

Numerical Gradient **Closed-form Gradient**

Figure 12: Quantitative comparison on the high-realism ShapeNet (without handpick: test case number 600 and 700). The reconstruction result of each approach is visualized in six different views, with the first view the same as the camera view, the first three views the same elevation as the camera view, and the last three view horizontal view.

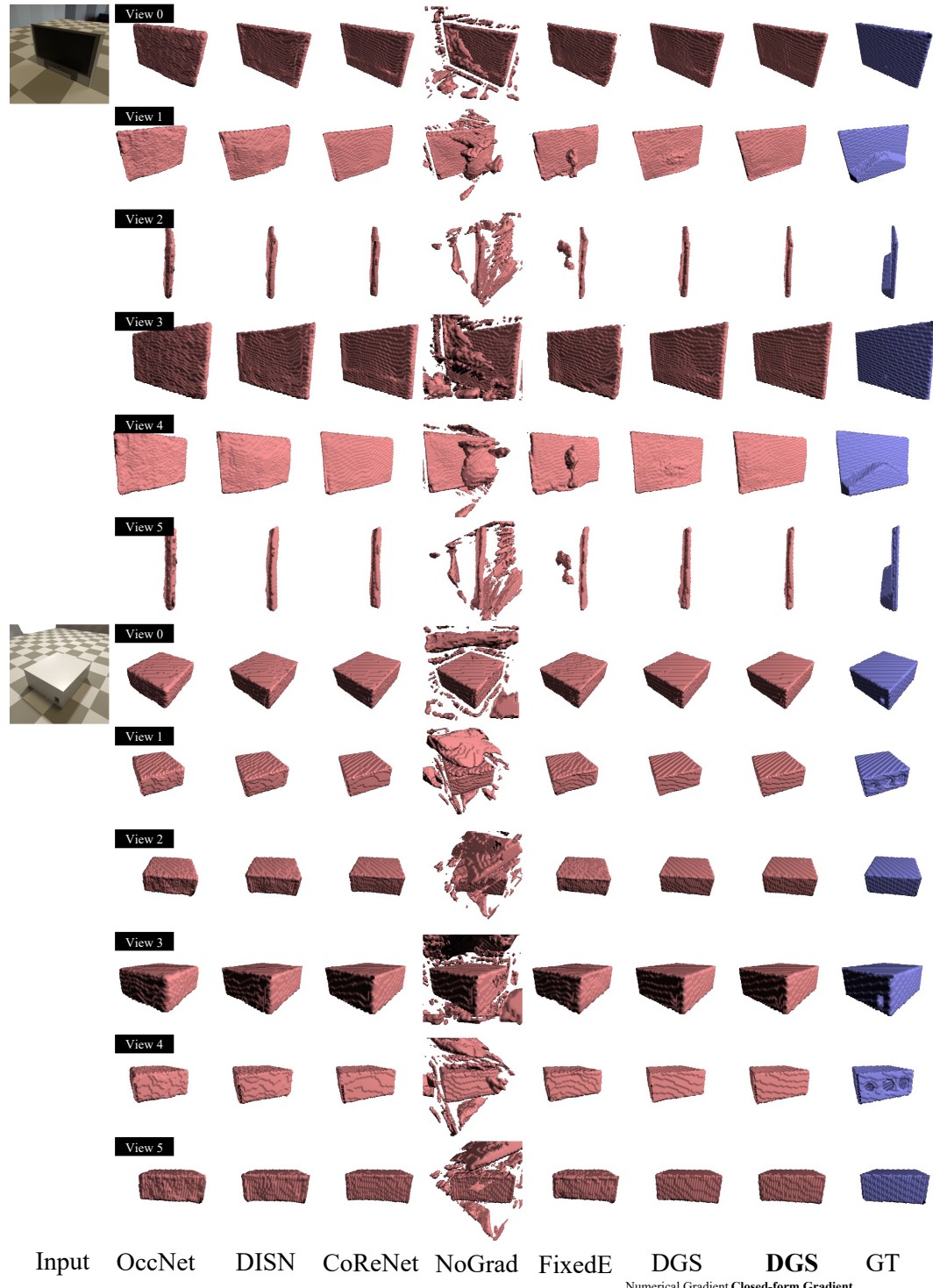

Figure 13: Quantitative comparison on the high-realism ShapeNet (without handpick: test case number 800 and 900). The reconstruction result of each approach is visualized in six different views, with the first view the same as the camera view, the first three views the same elevation as the camera view, and the last three view horizontal view.

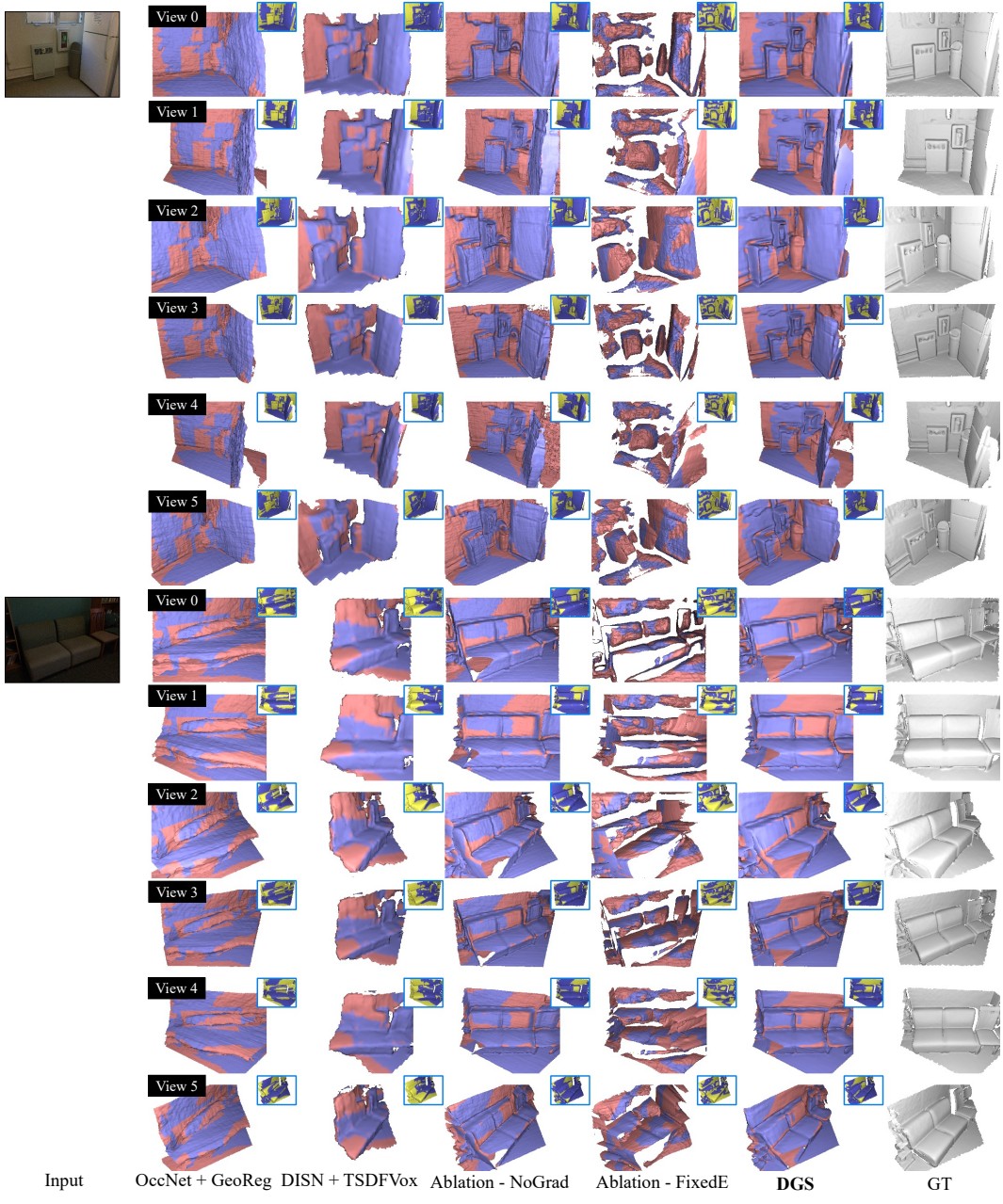

Input  OccNet + GeoReg DISN + TSDFVox Ablation - NoGrad Ablation - FixedE **DGS** GT

Figure 14: Quantitative comparison on the ScannetV2 (without handpick: the first frame of the 1st and 2nd test scene in ScannetV2). The reconstruction result of each approach is visualized in six different views, with the first view the same as the camera view, the first three views the same elevation as thecamera view, and the last three view elevated view.

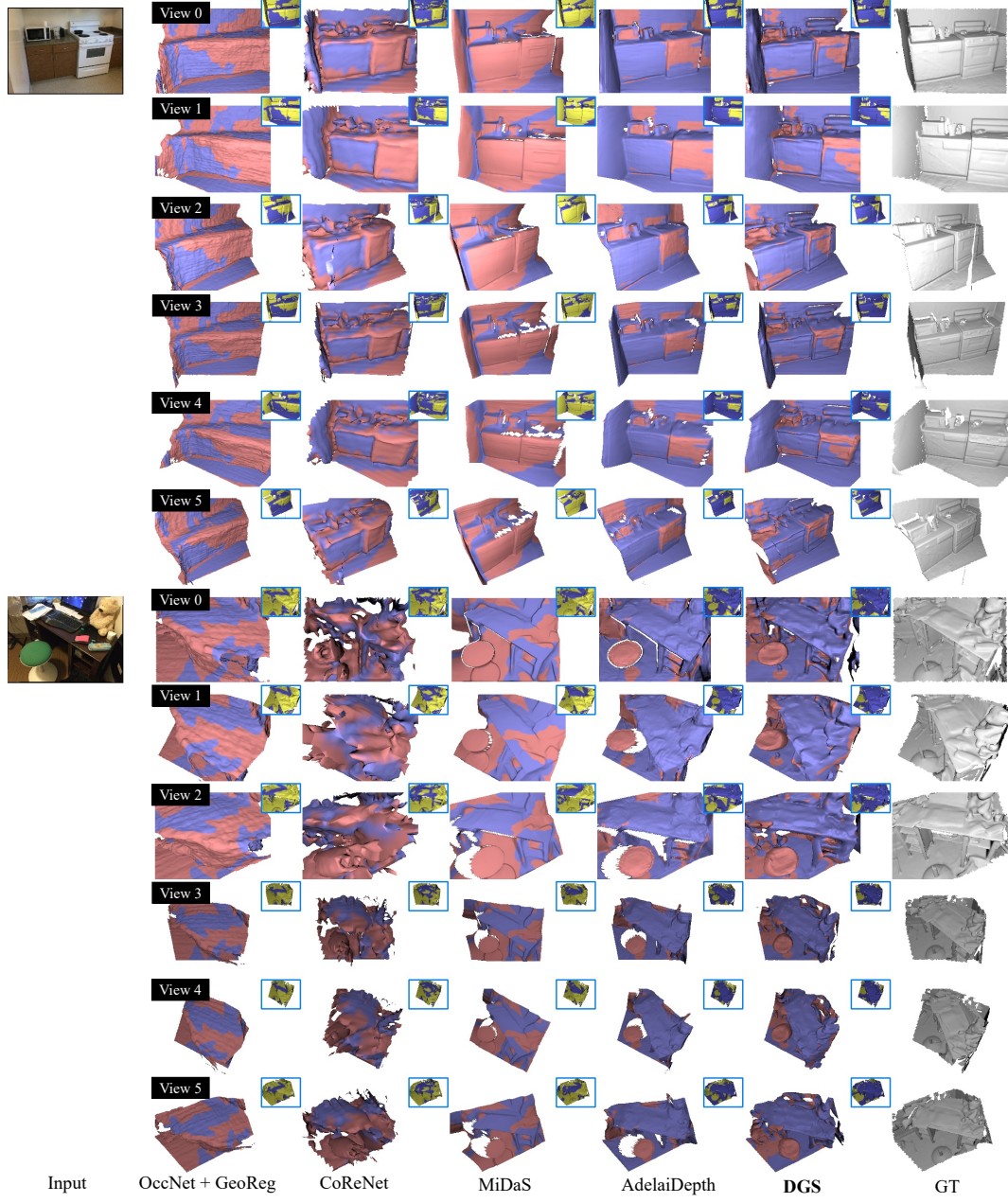

Input    OccNet + GeoReg    CoReNet    MiDaS    AdelaiDepth    **DGS**    GT

Figure 15: Quantitative comparison on the ScannetV2 (without handpick: the first frame of the 3rd and 4th test scene in ScannetV2). The reconstruction result of each approach is visualized in six different views, with the first view the same as the camera view, the first three views the same elevation as thecamera view, and the last three view elevated view.

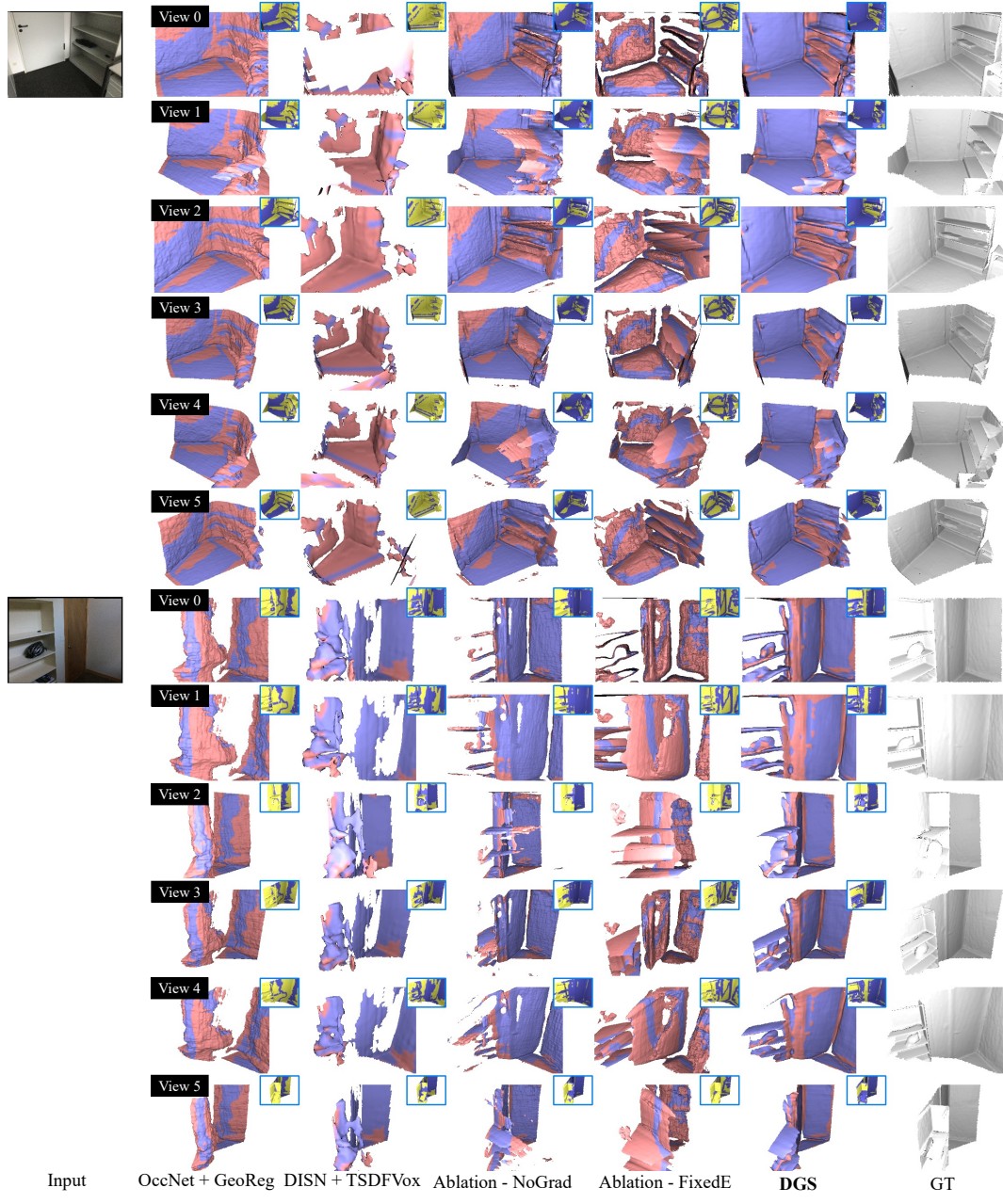

Input     OccNet + GeoReg  DISN + TSDFVox  Ablation - NoGrad   Ablation - FixedE   **DGS**     GT

Figure 16: Quantitative comparison on the ScannetV2 (without handpick: the first frame of the 5th and 6th test scene in ScannetV2). The reconstruction result of each approach is visualized in six different views, with the first view the same as the camera view, the first three views the same elevation as thecamera view, and the last three view elevated view.

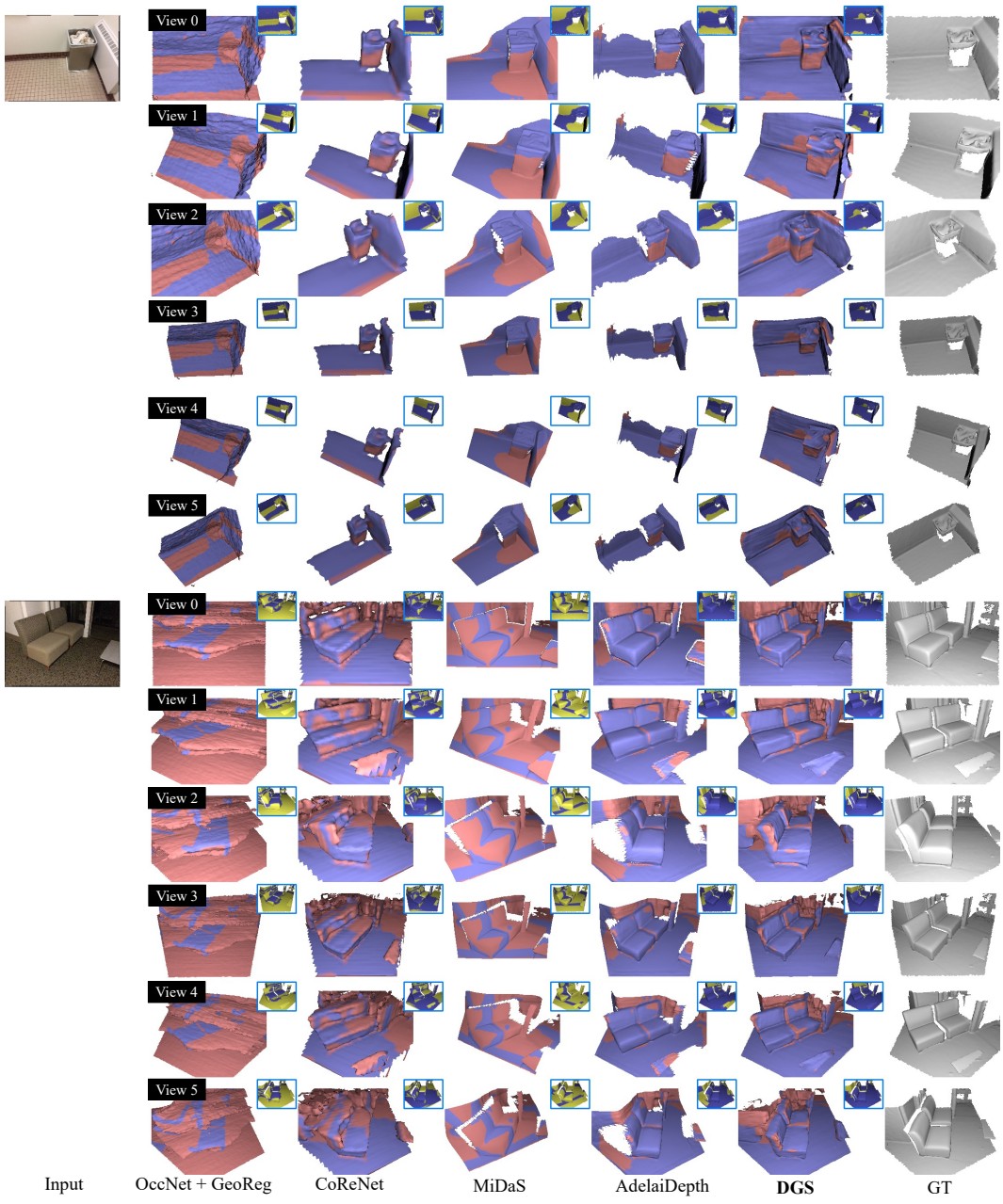

Figure 17: Quantitative comparison on the ScannetV2 (without handpick: the first frame of the 7th and 8th test scene in ScannetV2). The reconstruction result of each approach is visualized in six different views, with the first view the same as the camera view, the first three views the same elevation as thecamera view, and the last three view elevated view.

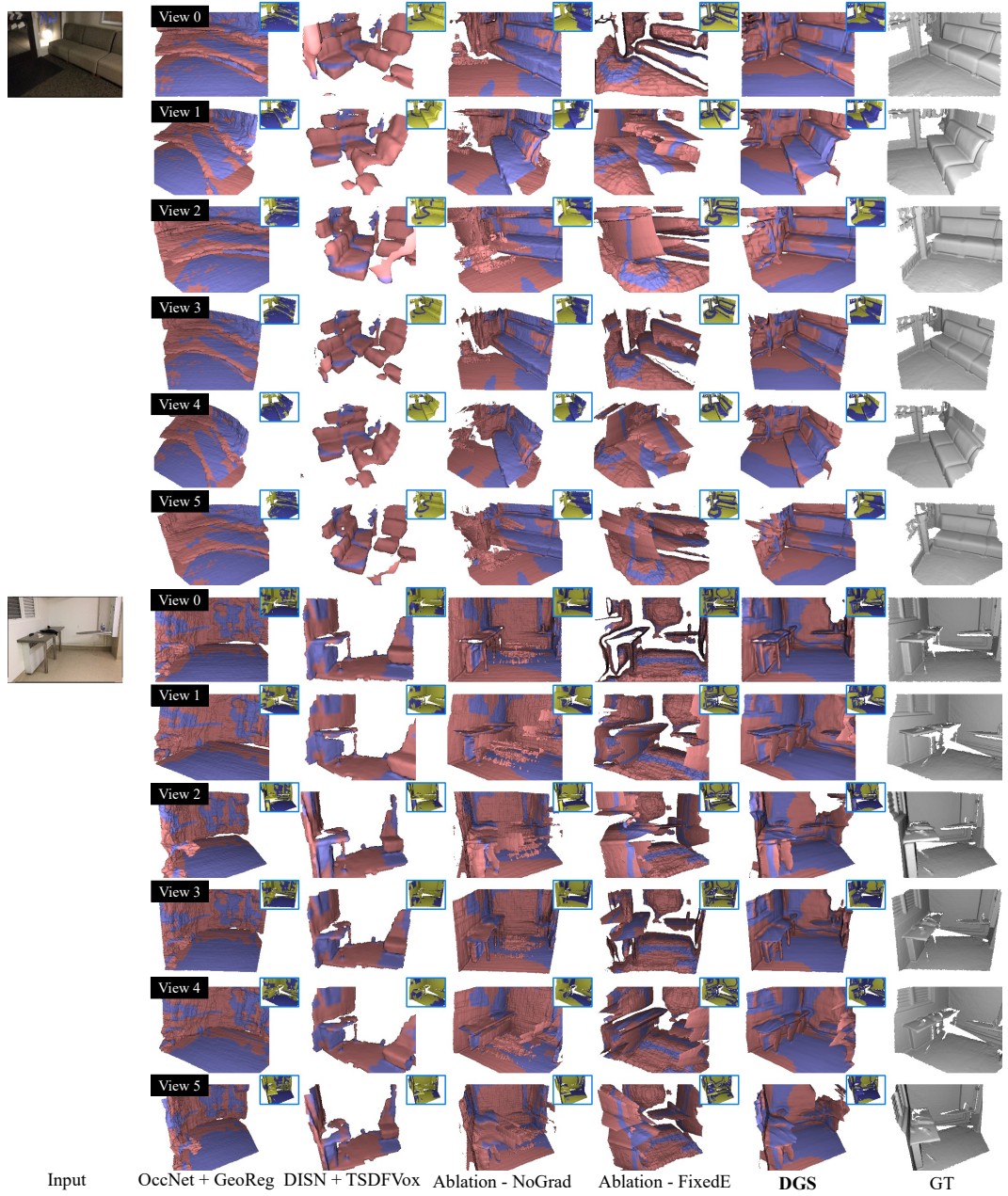

Input    OccNet + GeoReg   DISN + TSDFVox   Ablation - NoGrad    Ablation - FixedE    **DGS**    GT

Figure 18: Quantitative comparison on the ScannetV2 (without handpick: the first frame of the 9th and 10th test scene in ScannetV2). The reconstruction result of each approach is visualized in six different views, with the first view the same as the camera view, the first three views the same elevation as thecamera view, and the last three view elevated view.

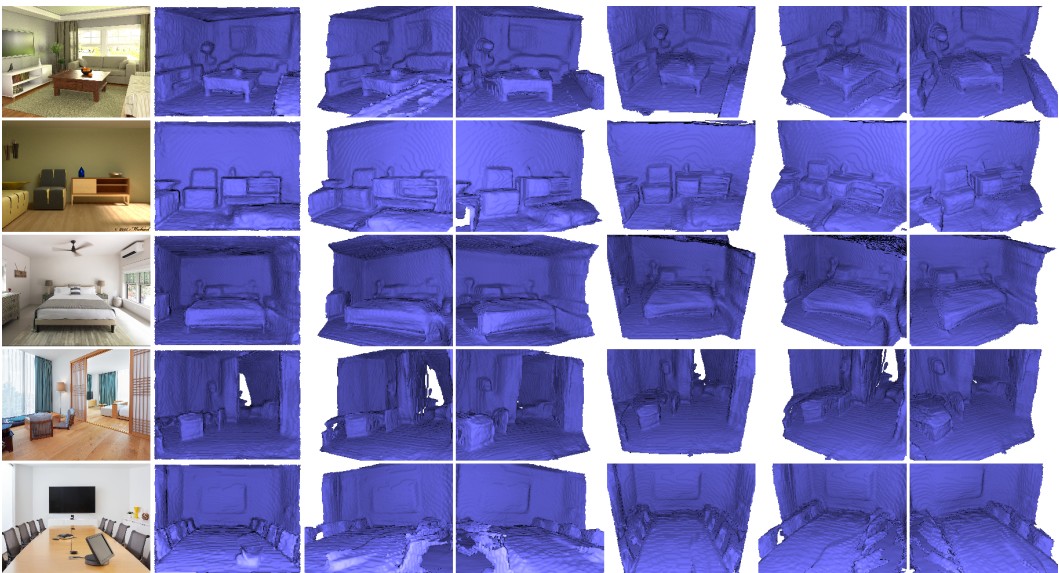

Figure 19: Additional Qualitative results of our model generalizing to unseen test images downloaded from the Internet.

