# OpenReview forum: "Differentiable Gradient Sampling for Learning Implicit 3D Scene Reconstructions from a Single Image"
_ICLR.cc/2022/Conference — ICLR 2022 Poster_

### Official Review · Reviewer_BEgr · 2021-11-01

**Correctness:** 3
**Technical Novelty And Significance:** 3
**Empirical Novelty And Significance:** 4
**Recommendation:** 6
**Confidence:** 4

**Details Of Ethics Concerns:**

/

**Main Review:**

## Strength
# Interesting and novel idea with the use of implicit representation
Implicit representation has been extensively explored in 3D object reconstruction and novel view synthesis recently. It is interesting to see how we can use implicit representation in various applications, e.g. single-view reconstruction.

# New sampling scheme for gradient
The authors have proposed a new sampling scheme for computing spatial gradient and thus a closed-form solution for loss propagation.
Although the approach is new, I still have some concerns listed below.

# Good result
The authors have shown good results in a variety of datasets, quantitatively and qualitatively.

# Adequate ablation study
An ablation study is also provided to support the effectiveness of the proposed method.


## Weakness
# Generalization evaluation
The authors provide a qualitative evaluation (single sample) on an unseen test image. It will be much better if more examples can be provided. Moreover, a quantitative evaluation will be more appreciated to show the generalization ability of the trained model.

# necessity of the gradient sampling scheme.
Though the proposed sampling scheme is new, I don't understand why it is necessary (i.e. eq 5). What will be the difference between this approach and a naive approach that compute a feature gradient map first, followed by simple differentiable sampling (eq.4).
Could authors shed some light on the difference? If the naive approach is a reasonable approach, why the proposed method is essential in this case?

**Summary Of The Paper:**

In this paper, the authors propose a new method for single view 3D reconstruction.
A conditional (image feature prior) implicit representation framework is proposed to reconstruct 3D scene from a single view.
In this paper, the authors propose that feature gradient is essential for watertight reconstruction and propose a differentiable gradient sampling method for the formulation.
Experiments have been performed on both synthetic and real datasets.
Superior results have been presented.

**Summary Of The Review:**

The paper is well written and presented overall.
The essential experiments are performed and the results are well presented.
However, I have a question regarding the major contribution (see weakness).

---

> ### Author Response · Authors · 2021-11-20
> **Response to Reviewer BEgr**
>
> We really appreciate the reviewer’s appreciation of our good results, novel learning representation and framework, as well as our adequate evaluation. Our approach provided the first attempt, to our knowledge, for unseen real-world single-view 3D implicit reconstruction, compared to existing works focusing on single objects or single scene fitting. We will release our full codes. We address the concerns below.
>
> __1. Generalization Evaluation.__
>
> We provide more qualitative results (images downloaded from the Internet) in Fig. 19 of the revision. Please kindly note that our Fig. 1 and Fig. 6 also show multiple unseen scene test images either downloaded from the Internet or from the Pix3D dataset (with the prediction visualized in the same deep blue color without ground truth). We also plan to add the quantitative generalization evaluation via testing our ScannetV2-trained model on Matterport3d. Since evaluating the whole test set takes time, we will update the revision again in the following week (__Update__: Please kindly note that the quantitative results are now available in Sec. E of our latest revision. Since our quantitative evaluation of a new testbed requires significant implementation during a limited time frame in the rebuttal period, we are confident but not 100\% sure about the correctness of Sec. E, given the short rebuttal period on new experiments - e.g. we are not sure why the current F1-score of DISN + TSDFVox is lower than both Prec. and Recall in Tab. 6. We will carefully revise Sec. E in the final version. Meanwhile, we ensure the correctness throughout the paper other than Sec. E).
>
> On the other hand, our quantitative evaluation on the ScannetV2 test set is also generalizing to unseen scenes. Our evaluation on other new test data mainly aims to rule out the dataset bias during testing. Compared to existing generalizable single-view approaches that are primarily trained on Shapenet objects with limited generalizability to the whole scene prediction of unseen real scenes, we believe our work demonstrates a significantly higher level of generalizability compared to existing approaches.
>
> __2. Necessity of the gradient sampling scheme.__
>
> We appreciate and thank the reviewer for suggesting this naive method. In our understanding, this can be implemented via obtaining the 2D gradient map, and then using Eq. 7 to compute the 3D spatial gradients. Practically we found that this would typically create two additional feature gradient maps, with the most memory intensive feature map whose map size 192 x 256, number of channel 256, and the batch size 32 in our settings. This inevitably brings difficulties for saving GPU memories during training (our training with the minimum batch size for good results would require ~40G of GPU memory). Our implementation instead processes everything on the fly, providing faster and memory efficient implementation that might facilitate future works that also use the feed-forward gradients for loss imposition.
>
> The contribution of our work also includes our good results, our novel learning framework with raw scan data, as well as our novel practical loss function (Eq. 1) that enables extending from the single object reconstruction or single scene fitting setting, in contrast to the existing Eikonal loss (Eq. 2). Please refer to Sec 3.1 of our revision for details.

---

### Official Review · Reviewer_BwV5 · 2021-11-01

**Correctness:** 3
**Technical Novelty And Significance:** 3
**Empirical Novelty And Significance:** 3
**Recommendation:** 6
**Confidence:** 4

**Main Review:**

pros:
1. the formation of DGS is novel and interesting, with promising performance.
2. detailed experiments on both shapenet and scan data
3. detailed ablation study
4. better performance compared with pervious methods.

cons:
1. the overall learning loss is not novel (Eq 1 & 2), for the Eikonal regularization part.
2. there are some camera  parameters are involved in DGS, which is a bit hard to get in general.
3. it is not clear what is the percentage of the know voxel occupancy/SDF,   and how much would the rate effect the learning. It ill be better to have some discussion and  ablations. IMHO, generally if the value of voxels closed to the surface is known, it might be ok to learn with other regions are missing.
4. from the results (Fig9 &10), DGS  seems to be more likely to generate floating points, some discussion would be better to have here.

**Summary Of The Paper:**

This paper presents a new method to learn implicit 3D scene reconstructions from single image input. The main improvement is a closed-form Differentiable Gradient Sampling.  By taking spatial gradient into  consideration, the proposed method can apply back-propagation of the  loss on spatial gradients to feature maps and allow the training for the case of without dense 3D supervision.

**Summary Of The Review:**

Overall, I think the proposed method is novel and with reasonable performance.  I am in favor for acceptance if the authors can provide some discussion about the cons listed above.

---

> ### Author Response · Authors · 2021-11-20
> **Response to Reviewer BwV5**
>
> We thank the reviewer for acknowledging our novelty and interesting ideas, promising performance and detailed evaluations. We demonstrated promising results for real-world scene 3D reconstruction. Compared to existing works that focused on objects or non-generalizable scene fitting, we believe our results and our learning framework would be important to the field. Full codes will be released. We address the concerns below.
>
> __1. The overall learning loss is not novel (Eq. 1 and 2) for the Eikonal part.__
>
> We claim that extending from the single objects or single scene fitting scenario to the generalizable real scene prediction setting is non-trivial, and we cannot naively apply existing techniques (e.g. Eikonal loss). We update the revision (Sec. 3.1) and make it clear that the Eikonal loss learning scheme would suffer from severe numerical difficulties and GPU memory issues so that it cannot reasonably train the model.
>
> __2. Camera parameters are involved in DGS, which is hard to get in general.__
>
> We noticed that it is the focal length (intrinsic parameters) that is coupled in the DGS formulation. Similar to the finding in (Popov et al. 2020, Sec 3.3 Obtaining camera parameters) that we can still obtain plausible 3D predictions even if the camera intrinsic parameters are unknown. In our experiments, all shapenet experiments are evaluated with the same focal length (FOV is 60 degrees). For ScannetV2, all the training images are cropped / resized so that the focal lengths are fixed. During testing (including on unseen images), we use this same focal length and get the plausible 3D prediction without the precise object scaling.
>
> __3. Percentage of known voxel occupancy affecting learning.__
>
> __(a) It is not clear what is the percentage of the known voxel … and how much would the rate affect the learning.__
>
> We provide the additional experimental results in the revision. In particular, as shown in Tab. 1, when enlarging the rate of known voxels from 10%, 30% to 50% via enlarging the near-surface margin, the performance of the model (without imposing our proposed gradient loss) increases steadily. This indicates it is important to know more voxel labels when our proposed gradient loss (Eq. 1) is not imposed. For real scenes, since we do not know the precise accurate occupancy labels for points far away from the surface, we cannot conduct similar experiments. However, based on the comparison between “Ablation-No Grad” and “DISN + TSDF Vox” (Tab. 2 of the revision, both using the same DISN architecture) we can see that the latter (learning with more known voxels from the inaccurate voxelization process) performs worse, which means knowing more inaccurate voxels deteriorate the results in the real scene scenario.
>
> __(b) If the value of the voxels close to the surface is known, it might be ok to learn with other regions missing.__
>
> Our benchmarking of the ablation-NoGrad baseline exactly evaluates this method, where the loss from only the near-surface query points are imposed for training. As shown in both the ShapeNet and ScannetV2 experiments, this baseline performs inferior compared to our approach quantitatively and qualitatively.
>
> __4. DGS seems to be more likely to generate floating points.__
>
> We notice that most of the floating points are right being in the occluded space. We noticed that most of the approaches evaluated in the paper do not explicitly model the occluded invisible region. In the two unhand-picked examples, our result indeed suffers more from this issue. We will leave the occlusion-invisible geometry modeling for future works.

---

### Official Review · Reviewer_SoBW · 2021-11-03

**Correctness:** 4
**Technical Novelty And Significance:** 3
**Empirical Novelty And Significance:** 4
**Recommendation:** 8
**Confidence:** 4

**Main Review:**

Their proposed loss functions are novel as far as I know.  The idea makes sense, to only apply supervision near surface boundaries where the labels can be reliably produced.  They perform a thorough set of experiments including comparisons and an ablation study.  They also derive the closed-form gradients of their loss function and show the importance of using them over numerical derivatives.  Both the quantitative and qualitative results are convincing.

One question I had was why there were no coefficients to balance the strength of the regularization terms in equations (1) and (2).

**Summary Of The Paper:**

This paper describes novel loss functions for learning to predict an implicit 3D scene representation from a single image.  They argue that when working with real scan data of scenes (rather than single objects) it is difficult to generate accurate occupancy or signed distance function (SDF) ground truth as would be required for supervised learning.  Instead, they propose to only use occupancy or SDF supervision near the surfaces of objects; elsewhere, they rely on constraints on the gradient of the occupancy or SDF adapted from Gropp et al. 2020.

They perform a thorough evaluation on several benchmark datasets and compare against state-of-the-art competing methods.  They show that they outperform competing methods, even though in some cases their method has access to less supervisory data.    They also perform an ablation study to show the importance of various parts of the loss function.



**Summary Of The Review:**

This paper has novel and interesting contributions to the field of single-image 3D reconstruction.  They provide convincing experiments to validate their contributions.  The paper is also well-written and nicely presented.

---

> ### Author Response · Authors · 2021-11-20
> **Response to Reviewer SoBW**
>
> We really appreciate the reviewer's appreciation of our results and our technical contribution. Our approach demonstrates the first attempt for implicit 3D single view reconstruction to unseen real scenes, achieving results that do not exist before. We believe it is important to the research field. Our novel learning framework with raw scan data as well as our novel occupancy gradient loss function (Eq. 1) and our time and memory efficient implementation of DGS might bring contributions for future works. Full codes will be released.
>
> We thank the reviewer for pointing out the issue on the loss balancing factor. We add the balancing factor for the loss terms in Eq. 1 and 2. In practice, we set the lambda_or in Eq. 1 to be 0.01 for the best result. We update these implementation details in the revision.

---

> > ### Comment · Reviewer_SoBW · 2021-11-29
> > **Response**
> >
> > Thanks very much for your response to the reviews and for addressing my comment about the loss balancing.  I have read the revised paper and other discussions and I will maintain my rating of accept.
> >
> > Some notes on the revised paper:
> > Figure 2: sufficintly -> sufficiently
> > occypancy -> occupancy
> > 3.1: occypancy -> occupancy
> > geometrical regularization -> geometric regularization
> > 3.2 "devised the model $f_\Theta$ fits to a single scene" -> this doesn't work grammatically and I am not sure what is trying to be expressed
> > 4.1 "how the rate of known voxels affect the learning performance" -> how the rate of known voxels affects the learning performance
> > 5 "that directly train" -> "that directly trains"

---

> > > ### Author Response · Authors · 2021-11-30
> > > **Many thanks for the careful check and suggestions**
> > >
> > > Many thanks for the detailed check on our rebuttal and the revision. We will address each in the manuscript, and will conduct the overall grammar check on the latest version to make sure our manuscript is carefully checked.

---

### Official Review · Reviewer_ZTzy · 2021-11-03

**Correctness:** 3
**Technical Novelty And Significance:** 3
**Empirical Novelty And Significance:** 3
**Recommendation:** 5
**Confidence:** 5

**Main Review:**

Strengths:
- I like the motivation of learning from 3D meshes that are not necessarily closed. This seems to allow 3D reconstruction neural networks to learn from a wider range of dataset resources, such as 3D scans or mesh reconstructions.
- The experimental settings are described in a very detailed manner with justifications. The results from the proposed method seem to show some improvements upon baseline methods, more notably in scene reconstructions.

Weaknesses:
- Although learning from either closed or open 3D meshes is an interesting motivation, this only allows one to learn SDF. One cannot define occupancy for open 3D meshes, and the example in Fig 3 is misleading. How would you define occupancy near open wholes? This part is unclear.
- The contribution is unclear and at most incremental. Two main components are described in the paper:
  1. Loss functions for learning occupancy or SDF (Sec 3.1). In my understanding, the only novel term is the spatial gradient penalty for occupancy. The rest of the two terms are standard loss functions for learning occupancy, so the additional first term is proposed. The "spatial gradient" term for SDF is precisely the eikonal regularization (Crandall & Lions, 1983), and the remaining terms are standard losses as well. (It is also unclear why the loss in the background paragraph is not incorporated).
  The authors argue that conditioning the spatial gradients on pixels is novel. I think the authors should clearly state why conditioning on pixels is novel enough to be a standalone paper itself, as this to me is an overstatement and I don't see how simply conditioning for a different task is novel. Penalty on the spatial gradients have also been previously adopted for other single-image 3D SDF reconstruction tasks [A,B]. Also, the authors mentioned that "the spatial gradient $\nabla_{x,y,z}\hat{f}_\Theta(x,y,z)$ can be conveniently computed without the sampling procedure" -- how?
  2. The gradient expression of "spatial gradients" (Sec 3.2). My understanding is that this is basically treating the finite pixel differencing operation as a network op, which in nature is differentiable. I think this part is confusing in many aspects. First, I don't see why the authors emphasize that the formulation is closed-form, as the spatial gradient expression is clearly taken numerically and its derivative can be computed via automatic differentiation. If the authors meant to present the gradient of gradient expressions, please use second-order derivatives (e.g. $\frac{\partial^2 f(x)}{\partial x^2}$). It is also unclear how the 3D case (Eq 7) is derived; the authors merely presented an equation without elaborating its meaning. It is unclear what exactly $h$ and $w$ are; the authors referred to Fig 4(c) but there are no explanations in the captions either, which prevents a total understanding. Finally, a very important reference of Spatial Transformer Networks [C] is missing, as it was the first to advocate differentiable sampling.
- It is unclear what "engineering constraints" refer to (Fig 2 caption and before Eq 1).
- What is the relationship between $x,y,z$ and $i,j$? These notations are cross-referenced throughout the paper but their distinction was never explained. In Fig 4, are the feature maps sampled according to $x,y,z$ or $i,j$? What does $\nabla_{x,y,z}\phi_{i,j}$ in Fig 4(a) mean?
- Experiments:
  - The results from the proposed method has its own training recipe (architecture, optimization etc), and thus it is unclear where the better performance is coming from. (It could not be about the losses or spatial gradients at all, but a better design of architecture.) I think it is essential to see results where the baseline methods (e.g. OccNet, DISN) are retrained with the proposed losses incorporated. Would these methods yield a boost of performance?
  - Why are different baseline methods compared for different versions (low-res and high-res) of ShapeNet? I think it's sufficient to present just the high-res version, and have a more complete comparison with the baselines in the low-res table.
  - Fig 5: it is undefined what "positive/negative precision/recall" mean. How close is a surface/surfel prediction to the ground truth is considered a "positive precision"? The red/blue figures are meaningless without precise definitions.
  - Could the authors elaborate more what "amodal depth" means, why the surface are evaluated in terms of this new metric, and how they are visualized? Why is not the naive depth definition used to evaluate? In addition, if only depth were used to evaluate quantitatively, why would one care about reconstructing the entire scene, as one could alternatively go for a scene depth prediction task which yields better quality?
- There is no conclusion section. What have we learned about this paper?

Other minor problems:
- Please use only a set of notations for occupancy labels, not both {0,1} and {+,-}. Also, please do not mix the use of $\phi(I)$ and $\phi$, where the former should be a function and the latter a variable.
- Fig 6: why is the bed in AdelaiDepth only visible in view 0?

[A] Jiang et al. "SDFDiff: Differentiable Rendering of Signed Distance Fields for 3D Shape Optimization." CVPR 2020.
[B] Lin et al. "SDF-SRN: Learning Signed Distance 3D Object Reconstruction from Static Images". NeurIPS 2020.
[C] Jaderberg et al. "Spatial Transformer Networks." NeurIPS 2015.


**Summary Of The Paper:**

This paper presents a method for 3D scene reconstruction from a single image using implicit surface representations such as occupancy or SDF. The authors propose to incorporate loss functions on the spatial gradients to provide dense supervision in the 3D space in the case where 3D labels may be incomplete (e.g. open 3D meshes) or not well-defined everywhere. Experiments are performed on ShapeNet and ScanNet show that the proposed method can achieve competitive performance on single-image scene reconstruction tasks.

**Summary Of The Review:**

I think this paper has an interesting motivation of learning 3D reconstruction from incomplete raw 3D scans / open meshes, but there are major flaws in the method description and the experiments (detailed above). I also don't think there are either sufficient novelty or insights in the paper. I think the submission needs much major revisions with additional experiments to validate the effectiveness of the proposed DSG.

---

> ### Author Response · Authors · 2021-11-20
> **Response to Reviewer ZTzy (5 / 5)**
>
> ### __Question 6__ - No conclusion?
>
> - We appreciate the suggestion and added it in the revision.
>
> &nbsp;
>
> ### __Question 7__ - Minor issues: Fig. 6 why is the bed in AdelaiDepth only visible in View 0?
>
> Only a small fraction of the bed depth surface is presented in View 1-3 due to view changes. We can see the surface of the far side of the bed (near the pillow) is not visible in AdelaiDepth, because there is little pixel in that region on the depth map. Those surfaces are filtered out because they are considered as "connecting two different depth layers". The uneven distribution of pixels for the same area of surface is also one of the reasons why depth prediction cannot fully replace occupancy prediction.

---

> > ### Comment · Reviewer_ZTzy · 2021-11-25
> > **Thanks for the response**
> >
> > I thank the authors for the very detailed response, and I appreciate the authors taking the effort to revise the submission. I do think the new version reads much smoother. Much of my confusion has been clarified, but some others still remain:
> > - I still think the presentation of Eq 7 is abrupt. I think the authors should at least elaborate how Eq 7 is derived (e.g. from the 2D case and/or plugging in some other expressions), especially since this is just as important as the 2D case. Even one or two intermediate steps would be helpful, but the current form is still not straightforward to comprehend.
> > - Positive/negative P/R: I think it's necessary to explicitly restate the definitions, and this doesn't seem to have been incorporated in the revision. It's not a problem to borrow metrics from previous works, but these are not standard metrics and I don't think it should be assumed that such metrics are commonly known.
> >
> > I am willing to raise my rating after reading the revised submission and the rebuttal.

---

> > > ### Author Response · Authors · 2021-11-26
> > > **Many thanks for the comments and score raising**
> > >
> > > We thank the reviewer for taking time helping to check our revision and responses as well as the suggestions to further improve the manuscript. We really appreciate the reviewer’s appreciation of our presentation and the ideas in the revision. For Eq. 7, we will provide the expansion of the chain rule as the intermediate steps (details below). For the evaluation metrics, we will add the definition and the details (details below) and move part of the additional details in the same paragraph to the appendix. We will carefully address these updates.
> > >
> > > -------------------------------------------------------------------
> > >
> > > __Updated Eq. 7__
> > >
> > > $ \frac{\partial \phi(x, y, z)}{\partial x} = \frac{\partial \phi(i, j)}{\partial i} \cdot \frac{\partial i}{\partial x} + \frac{\partial \phi(i, j)}{\partial j} \cdot \frac{\partial j}{\partial x} = \frac{\partial \phi(i, j)}{\partial j} \cdot \frac{f_0}{z}$
> > >
> > > $ \frac{\partial \phi(x, y, z)}{\partial y} = \frac{\partial \phi(i, j)}{\partial i} \cdot \frac{\partial i}{\partial y} + \frac{\partial \phi(i, j)}{\partial j} \cdot \frac{\partial j}{\partial y} = \frac{\partial \phi(i, j)}{\partial i} \cdot \frac{f_0}{z} $
> > >
> > > $  \frac{\partial \phi(x, y, z)}{\partial z} = \frac{\partial \phi(i, j)}{\partial i} \cdot \frac{\partial i}{\partial z} + \frac{\partial \phi(i, j)}{\partial j} \cdot \frac{\partial j}{\partial z} = \frac{\partial \phi(i, j)}{\partial i} \cdot (-\frac{yf_0}{z^2})
> > >                                                + \frac{\partial \phi(i, j)}{\partial j} \cdot (-\frac{xf_0}{z^2}) $
> > >
> > > __Updated evaluation metrics definition and details (to be added to Sec. 4.2 - after the first sentence of the “metric” paragraph)__
> > >
> > > In particular, the metrics evaluate the discrepancy between the predicted surface with the ground truth surface. Accuracy (Acc) denotes the average projecting distance from the predicted surface to the ground truth surface, and Completeness (Compl) denotes that from the ground truth surface to the predicted one. Chamfer Distance (CD) is their arithmetic mean. Precision (Prec) denotes the percentage of the predicted surface whose projection distance to the ground truth surface is below a threshold (0.05 meter following Sun et al. 2021; Murez et al. 2020), and Recall denotes that when projecting the ground truth surface to the predicted one. F1 score is their harmonic mean for each test case. We mainly evaluate the surface because obtaining the accurate per point occupancy label is not easy given the non-watertight meshes. Evaluating the surface also enables our comparison to the depth prediction approaches (Ranftl et al. 2019; Yin et al., 2021).

---

> ### Author Response · Authors · 2021-11-20
> **Response to Reviewer ZTzy (4 / 5)**
>
> ### __Question 5__ - Experiments (Continued)
>
> &nbsp;
>
> __(2) Low and high realism shapenet.__
>
> We use low-realism Shapenet as the majority of the supervised single view implicit surface reconstruction approaches reported the performance on this set, and hence we can directly compare with the numbers in those literatures. On high-realism shapenet, we retrain all the approaches and compare them with exactly the same settings (captions of Table 1 of the revision). We appended a new row in the revision as D2Im-Net's code was released very recently.
>
> &nbsp;
>
> __(3) Fig 5: it is undefined what "positive/negative precision/recall" means.__
>
> Our evaluation metrics strictly follow Sun et al. (2021) and Murez et al. (2020) (the first line of the metrics paragraph in Sec. 4.2), and so this includes the definition of "positive/negative precision/recall". Precisions correspond to the point cloud extracted from the predicted surface, where the positive precision means the points whose projection distance to the ground truth surface is smaller than 0.05 meters. Recalls correspond to the point cloud extracted from the ground truth surface, where the positive recall means the points whose projection distance to the predicted surface is smaller than 0.05 meters. The higher the better for both the metrics.
>
> &nbsp;
>
> __(4) amodal depths.__
>
> __(a) Could the authors elaborate more what "amodal depth" means.__ In our 3D setting, the amodal depth means the depth map of the scene constructed by only the layout surfaces, e.g. the wall, the ceiling, the floor etc without all other scene objects. It is mainly used for delineating our space of interest inside the camera view frustum - e.g. we do not predict the 3D surfaces behind the wall - we only predict the space in front of the wall.
>
> __(b) Why are the surfaces evaluated in terms of this new metric, and how are they visualized? Why is not the naive depth definition used to evaluate?__
>
> Please kindly note that we do not use new metrics in our evaluation, and in particular we do not evaluate amodal depth - it is just for delineating the evaluation scope. Our evaluation of depth is naive depth.
>
> __(c) Why not only predict the depth?__
>
> For three reasons.
>
> First, depth prediction only predicts visible surfaces, and it cannot predict per-point occupancy, specifically the points being occluded by nearer surfaces. It limits the modeling of the geometry of objects (including the backside), inter-object relations or affordances. It also limits downstream tasks - e.g. the free space estimation for planning in robotic navigation, and novel view hallucination.
>
> Second, for the visible depth surface, the pixels might be unevenly distributed for the same area of the surface, which will lead to the wrong estimate of the surface prediction. For example, the AdelaiDepth result in Fig. 6, there is very little depth pixel representing the far side of the bed (near the pillow). Those surfaces are filtered out because they are considered as "connecting two different depth layers". We observe that its F1 score would be much worse if we do not filter out pixels “connecting two different depth layers”.
>
> Third - incomplete prediction of the pixel map. We also noticed from the results of the AdelaiDpeth that sometimes a considerable amount of the depth pixels are marked as "not predicted" (e.g. Fig. 17). Instead, our predicted occupancy field for all the points within the view frustum can lead to 100\% completeness.

---

> ### Author Response · Authors · 2021-11-20
> **Response to Reviewer ZTzy (3 / 5)**
>
>
>
> ### __Question 3__ - What is the engineering constraint?
>
> We revised the caption of Fig 2 and the sentence before Eq. 1. The engineering constraint is referring to the difficulties of storing and run-time loading the sufficiently dense pre-computed occupancy labels for all the samples in the batch (batch size is 32) when training with the pre-computed full occupancy labels (our baseline of DISN + TSDF Voxelization), as we introduced in the introduction section. Hence for this baseline, we train with the relatively sparse labels (the voxels), whose resolution is lower than the pre-computed dense occupancy labels.
>
> &nbsp;
>
>
> ### __Question 4__ - $x, y, z$ and $i, j$
>
> __(1) What is the relationship between $x, y, z$ and $i, j$?__
>
> $i, j$ are the 2D projection from the 3D camera coordinate system to the 2D pixel coordinate system (the normalized device coordinate NDC used by sampling). We added their definition and relation before Eq. 3 and Eq. 7 in the revision.
>
> &nbsp;
>
> __(2) In Fig 4, are the feature maps sampled according to $x, y, z$ or $i, j$?__
>
> They are according to $i, j$. $x, y, z$ are the 3D camera coordinates.
>
> &nbsp;
>
> **(3) What does $\nabla_{x, y, z}\phi_{i, j}$ mean?**
>
> It is the 3D spatial gradient of the feature value $\phi_{i,j}$ sampled from the pixel location $i, j$. Note $i$ and $j$ are continuous NDC coordinates. We point out that it can also be written as $\nabla_{x, y, z}\phi_{x, y, z}$. we put the $i,j$ or $x,y,z$ as the subscript of $\phi$ just meaning that this is the sampled feature value, rather than the pre-sampled feature value on the raw feature map (e.g. $\phi_A, \phi_B, \phi_C, \phi_D$).
>
> &nbsp;
>
>
> ### __Question 5__ - Experiments
>
> __(1) "It is unclear where the better performance is coming from". "OccNet, DISN with the proposed loss".__
>
> As presented in our submission, in our shapenet experiments (e.g. the captions of Table 1 in the revision), DGS is based on CoReNet architecture, and we maintain the same setups - the only difference is the loss we used. We did not use any privileged settings during comparison - e.g., we found doubling the batch size of CoReNet from 4 to 8 would yield a significant boost (64.9 as mean IoU), and the higher the batch size the better result will be - and we did not use it in our experiments. Our ScannetV2 experiments are also fair comparison - all the models are trained with the same encoders, optimizers and learning rate settings etc.
>
> We further experimented with trying OccNet and DISN with the proposed novel loss and reported the performance in Tab. 1 of the revision. In particular, DISN w/ Eq. 1 requires DGS. We can see OccNet w/ Eq. 1, DISN w/ Eq. 1 and DGS (CoReNet w/ Eq. 1) consistently outperform OccNet, DISN, CoReNet in the majority of the categories.
>
> It is one of the interesting findings of this work that our approach prevails in our eye-to-eye comparison with training with the full supervision on per query points. As presented in the submission, we attribute our superior performance to our particular focus toward the difficult close-to-surface region similar to hard negative mining, as the majority of our sampled query points are close-to-surface (Sec 3.1 in the revision). Although our performance converges slower among the training iterations, it finally converges to a better number.

---

> ### Author Response · Authors · 2021-11-20
> **Response to Reviewer ZTzy (2 / 5)**
>
> ###  __Question 2__ - Technical Contribution (Continued)
>
> &nbsp;
>
> __(1) Novelties on the loss function (Sec 3.1).__
>
> __(a) Losses Functions in Eq. 1.__ We believe the three terms as a whole in Eq. 1 form a novel learning framework that enables learning from non-watertight meshes for the first time. To our knowledge, no existing work trains with occupancy labels of near-surface points only.
>
> __(b) Why is the loss in the background paragraph not incorporated?__ The loss in the background is incorporated partially for both the occupancy (Eq. 1) and the SDF (Eq. 2) scenarios - For Eq. 1, we only know the near-surface occupancy (the last two terms); For Eq. 2, we only know the SDF value of the surface point to be 0 (the second term).
>
> __(c) The authors should state clearly why conditioning on pixels is novel enough to be standalone paper itself.__ We respectfully point out that we did not claim "conditioning the spatial gradients on pixels" alone is novel. Instead, this "conditioning" poses non-trivial difficulties on back-propagating the loss gradient of the spatial gradient over the sampling module (Eq. 8). This makes the "conditioning" distinguished from existing literature. In addition, our contributions also include the good results, the overall learning framework that learns from raw scan data, as well as our novel occupancy gradient loss (Eq. 1).
>
> __(d) Gradient penalty has also been adopted in other single image tasks.__ Existing works [A, B] do not face the difficulties because [A, B] use global features for predictions, without requiring the local feature sampling procedures. It is important to extract local features when extending from single objects [A, B] to real scenes (see visual results of OccNet + GeoReg v.s. DGS). We added the reference and the discussion in the revision.
>
> __(e) “The spatial gradient can be conveniently computed without the sampling procedure.” How?__ We rephrase this sentence in the revision - this is not our approach. Rather, this is the GeoReg paper (Gropp et al. 2020) that fits a model to a single scene where autograd can achieve that - the spatial gradient can be conveniently computed because it does not involve the sampling procedure. In our generalizable feed-forward prediction case, it instead becomes a difficulty, because now the spatial gradient computation must undergo the sampling procedure. This further demonstrates the value of our proposed learning framework.
>
> &nbsp;
>
> __(2) The gradients (Sec 3.2).__
>
> __(a) "This is basically treating the finite pixel differencing operation as a network op, which in nature is differentiable".__ We found this operation cannot be directly achieved using the Autograd package of PyTorch for the grid_sample function during back-propagation. Our cuda implementation of this differentiable layer processes all the data on-the-fly, demonstrating advantages of time and memory efficiency and providing essential support to our learning system. The implementation is non-trivial, and we will release it along with the full code to the public. We believe this is an important contribution to the field and will facilitate many follow-up works for back propagation losses over the feed-forward gradients.
>
> __(b) "I don't see why the authors emphasize that the formulation is closed-form".__ Our claimed closed-form is relative to the numerical approximation counterpart (Sec C) where a $\delta$ is set for perturbing each query point for computing the gradient in the numerical approximation way. Empirically we found that, in the numerical counterpart, large $\delta$ leads to inaccurate gradient computation, while small $\delta$ leads to numerical difficulties when propagating through the deep network. Our closed-form solution is the $\delta$=0 solution without the numerical concerns. Our results in Sec C indicate our numerically stable closed-form solution achieved better results compared to the numerical counterpart.
>
> __(c) "How Eq.7 derived ... what exactly $h$ and $w$ are ... missing reference [C]".__ $h$ and $w$ are the height and the width of a single pixel within the normalized device coordinate (the same coordinate system used by pixel map sampling). We supplement these details and the reference in the revision.

---

> ### Author Response · Authors · 2021-11-20
> **Response to Reviewer ZTzy (1 / 5)**
>
> We thank Reviewer ZTzy for raising all the detailed issues for us to clarify to make sure our contributions are well conveyed and acknowledged by all the readers. We appreciate that the reviewer likes our motivation to learn directly from the raw scan data for widening training data sources, as well as the novelty of our occupancy loss.
>
> __Please kindly refer to "Authors Common Response" for our summarized contributions.__
>
> &nbsp;
>
> ### __Question 1__ - Occupancy loss for open meshes
>
> __(1) The gradient losses only allow one to learn SDF. Figure 3 is misleading.__
>
> We now have clarified and made this clear in the revision (Sec. 3.1). Our novel occupancy gradient loss (Eq. 1, Fig. 3c) is our dominant learning scheme throughout our experiments. In short, the Eikonal term (Eq. 2) suffers from numerical difficulties and GPU memory bottleneck and cannot be easily applied to our unseen real world scenario.
>
> __(2) How would you define occupancy near open holes?__
>
> In the real world, the occupancy is determined by the physical occupancy. The raw scan data introduces the noise (the holes) and the occupancy cannot be determined solely based on that. Our occupancy losses (Fig. 3) recovers the occupancy via favouring the minimum area of hole filling - in the hole region, the "occupied" and "unoccupied" space are connected, and to minimize the gradient loss, our model is seeking for the minimum area of hole filling - so achieving the minimum number of the sampled points that have non-zero occupancy gradients. We point out that hole filling is an under-determined, multi-solution problem, and our loss provides one reasonable solution for training.
>
> &nbsp;
>
> ### __Question 2__ - Technical Contribution
>
> __(0) The contribution is unclear and at most incremental.__
>
> We respectfully disagree.
>
> First, we believe our contributions are beyond just maths. As acknowledged by other reviewers, our "good results" "do not exist in prior works". We provide the very first attempt to propose to learn real scene single view 3D implicit reconstruction via training directly from raw scan data. This opens up the potential for accurate single view geometry reconstruction for real images.
>
> Second, naively applying existing approaches (E.g. the Eikonal term Eq. 2 or all the baseline approaches in Tab. 1 of the revision and Fig. 5, 6) to our unseen real world scenario would demonstrate difficulties and cannot achieve reasonable results. Thanks to our novel learning framework to train directly from raw scan data, our novel occupancy gradient loss (Eq. 1) as well as our time and memory efficient implementation of the differentiable gradient sampling layer, we enable good results that “do not exist in prior works”.

---

### Author Response · Authors · 2021-11-20
**Authors’ Common Response**

We thank all the reviewers for the constructive comments. We provide a summary of our contribution as well as the checklist of revisions here.

&nbsp;

### __Summary of Contributions__

__"Good results" that "do not exist in prior works"__

We provide good results (e.g. Fig. 1) as the first attempt for unseen real scenes single view implicit 3D reconstruction. The extension from single scene fitting or single objects reconstruction to unseen real scene is non-trivial and has not been done before. As acknowledged by Reviewer SoBW, BEgr and BwV5, our empirical novelty and contributions "are significant, and do not exist in prior works" (Reviewer SoBW and BEgr) "with promising performance" (Reviewer BwV5).

Existing neural field single-view prediction approaches mainly focused on objects. We push this boundary with the proposed learning framework that is necessary to train on real scenes. We believe our "good results" contribution alone would open up the potential for accurate real scene single view implicit 3D reconstructions and is important to this field. Full codes will be released.

__Technical contribution - "Novel and interesting ideas"__

We show that naively applying existing approaches cannot achieve our novel results. We list each methodology novelty contribution below:

(1) Our novel occupancy gradient loss (Eq. 1, Fig. 3c) is our dominant learning scheme in all our experiments. We now have made it clear in the revision that the Eikonal term (Eq. 2) suffers from numerical difficulties and GPU memory bottleneck, and is not easy to be used in our unseen real world scenarios.

(2) We propose to directly learn from the imperfect raw scan data with our overall novel learning framework, and thus open up the potential for accurate high resolution single view 3D reconstruction.

(3) Our cuda implementation of the differentiable gradient sampling layer provides a time and memory efficient solution for our particular task. It will also bring important benefits for future works that impose the loss toward any feedforward gradients.

__"Adequate evaluation"__

As acknowledged by Reviewer SoBW, BEgr and BwV5, we "performed a thorough evaluation", "compared against state-of-the-art competing methods" and "achieved superior results". We "provided convincing experiments to validate our contributions". We also conducted "adequate" "detailed" "ablation study" to "support the effectiveness" and "show the importance of various parts of the loss function". "Both the quantitative and qualitative results are convincing."

__"Well written and nicely presented"__

As acknowledged by Reviewer SoBW and BEgr, "The paper is also well-written and nicely presented"; "The paper is well written and presented overall".


&nbsp;

### __Checklist of Revisions__

Please kindly check our rebuttal revision. We have marked all the contents updated (not including minor changes e.g. typo fixing or text order swapping) with the red color. In addition to the red text indications, we also made the following changes when compared to the original submission:

(1) We deleted the last paragraph of the related works;

(2) We move the original Tab. 1 (low-realism ShapeNet experimental comparisons) to the appendix following Reviewer ZTzy’s suggestion.

(3) We move the metrics, implementation details of our approach from Sec. 4 to the appendix (Sec. A and B).

---

### Decision · Program_Chairs · 2022-01-20

**Decision:**

Accept (Poster)

**Comment:**

The paper presents a new way to train the prediction of implicit 3D scene representations from a single view. The main innovations are a novel numerically stable and memory efficient formulation of the derivatives of a loss function based on the spatial gradients of the implicit field, and focusing the training on regions near the surfaces of objects. The method leads to good performance, especially when training on imperfect ground truth scan data.

Concerns were raised about the novelty of the approach and its significance. These were adequately addressed in the author response and revisions. The experiments were found to be well described and executed, which increases the confidence in the approach and its potential impact. I recommend acceptance.